# Gene-editing in patient and humanized-mice primary muscle stem cells rescues dysferlin expression in dysferlin-deficient muscular dystrophy

Helena Escobar [1,2,3] ✉, Silvia Di Francescantonio[1,2,3], Julia Smirnova[1], Robin Graf [1,2,3], Stefanie Müthel [1,2,3], Andreas Marg [1,2,3], Alexej Zhogov[1,2,3], Supriya Krishna[1,2,3,4], Eric Metzler[1,2,3], Mina Petkova[5], Oliver Daumke [1,4], Ralf Kühn [1] & Simone Spuler [1,2,3,4] ✉

Dystrophy-associated fer-1-like protein (dysferlin) conducts plasma membrane repair. Mutations in the *DYSF* gene cause a panoply of genetic muscular dystrophies. We targeted a frequent loss-of-function, *DYSF* exon 44, founder frameshift mutation with mRNA-mediated delivery of SpCas9 in combination with a mutation-specific sgRNA to primary muscle stem cells from two homozygous patients. We observed a consistent >60% exon 44 re-framing, rescuing a full-length and functional dysferlin protein. A new mouse model harboring a humanized *Dysf* exon 44 with the founder mutation, hEx44mut, recapitulates the patients' phenotype and an identical re-framing outcome in primary muscle stem cells. Finally, gene-edited murine primary muscle stem-cells are able to regenerate muscle and rescue dysferlin when transplanted back into hEx44mut hosts. These findings are the first to show that a CRISPR-mediated therapy can ameliorate dysferlin deficiency. We suggest that gene-edited primary muscle stem cells could exhibit utility, not only in treating dysferlin deficiency syndromes, but also perhaps other forms of muscular dystrophy.

Loss-of-function mutations in *DYSF*, encoding dysferlin, cause autosomal recessive muscular dystrophy syndromes, notably limb-girdle (LGMD2B/R2)[1,2]. The *DYSF* gene spans >150 kb of genomic DNA in human chromosome 2p13. Dysferlin is a ~240 kDa protein resulting from a >6 kb coding sequence comprising 55 canonical exons[3,4]. It anchors to cell membranes and intracellular vesicles by a short single-pass transmembrane domain that has a very small C-terminal extracellular domain. The cytosolic part contains seven C2 domains (C2A to C2G) that bind phospholipids in a calcium-dependent manner[5–7]. Dysferlin repairs muscle-fiber membranes (sarcolemma), where it mediates calcium-dependent vesicle recruitment to the site of injury[8]. It has also been implicated in calcium homeostasis and lipid metabolism[9,10]. Dysferlin deficiency leads to progressive skeletal muscle degeneration and atrophy[11,12]. Restoring dysferlin expression is likely the only curative solution and clustered regularly interspaced short palindromic repeats–associated protein 9 (CRISPR/Cas9) methodology would have the greatest utility.

[1]Max Delbrück Center for Molecular Medicine in the Helmholtz Association (MDC), Berlin, Germany. [2]Charité—Universitätsmedizin Berlin, Corporate Member of Freie Universität Berlin and Humboldt-Universität zu Berlin, Charité Campus Buch, Berlin, Germany. [3]Muscle Research Unit, Experimental and Clinical Research Center (ECRC), a joint cooperation between the Charité—Universitätsmedizin Berlin and the Max Delbrück Center for Molecular Medicine in the Helmholtz Association (MDC), Berlin, Germany. [4]Department of Biology, Chemistry, and Pharmacy, Freie Universität Berlin, Berlin, Germany. [5]MyoPax GmbH, Berlin, Germany. ✉e-mail: helena.escobar@mdc-berlin.de; simone.spuler@charite.de

CRISPR/Cas systems are currently the most promising and effective genetic tools to site-specifically edit mammalian genomes and thereby repair gene defects that cause human disease[13,14]. Most cells repair Cas9-induced DNA double-strand breaks (DSB) preferentially through non-homologous end joining (NHEJ) pathways, whereby the free DNA ends are re-ligated somewhat imprecisely, usually leading to the introduction of small insertions or deletions, so-called indels[14]. At some sites, Cas9 preferentially induces staggered-end cleavage, often resulting in one-nucleotide overhangs that are re-filled by DNA polymerases. This process typically leads to a duplication of the fourth base pair 5′ upstream the protospacer adjacent motif in a non-random fashion[15–18]. Cas9-mediated NHEJ with single or dual guide RNAs has very effectively been used to inactivate splice sites or induce deletions of single or multiple exons in the dystrophin-encoding *DMD* gene in muscle cells, so as to induce skipping of exons with the most frequent Duchenne's muscular dystrophy–(DMD) causing mutations[19]. This strategy can rescue non-sense and frameshift mutations when the skipped protein domains are to some extent dispensable, as is the case for dystrophin. However, in LGMD2B, functionally redundant domains in dysferlin and mutational hotspots are lacking[20]. Nonetheless, there are a few *DYSF* founder mutations with a high prevalence in some population subsets.

Muscle homeostasis and regeneration rely on muscle stem cells (MuSC), a tissue-specific pool of adult stem cells located adjacent to multinucleated muscle fibers. After injury, they proliferate extensively and give rise to numerous myogenic progenitor cells. These cells can further differentiate and fuse to existing fibers, or to one another, to generate new fibers. They can also repopulate the stem-cell niche to maintain the MuSC pool long-term[21]. These cells are an attractive population for cell replacement therapies; however, obtaining pure MuSC from patients has been challenging due to, among others, fibro-fatty replacement of muscle tissue[22,23]. We recently succeeded in isolating pure MuSC populations from an α-sarcoglycan-deficient patient with a G > A mutation that we were able to efficiently repair by base editing. We showed that base-edited patient MuSC regenerated muscle and repopulated the stem-cell compartment in xenografts[24]. Furthermore, we showed that mRNA is a well-suited substrate to transiently deliver gene editing tools to primary human MuSC, systematically enabling very high editing rates, without any selection steps. Despite interventions, the myogenic properties of the cells were maitainted[25,26].

Here, we sought to harness Cas9-mediated NHEJ to rescue a founder frameshift mutation in *DYSF* exon 44 (c.4872_4876delinsCCCC)[27] without skipping any exons. We identified a mutation-specific guide RNA (gRNA) that, in combination with SpCas9, introduced a templated +1A nucleotide insertion at the DSB in human induced pluripotent stem cells (hiPSC) from two homozygous patients. Delivery of SpCas9 mRNA, plus the gRNA, resulted in >90% on-target editing with >60% +1A insertion rates in primary patient MuSC. Similar utility was observed in MuSC from a novel mouse model carrying the mutant human *DYSF* exon 44 in the endogenous mouse *Dysf* locus, hEx44mut. Editing rescued the reading frame and restored full-length dysferlin protein in patient and mouse MuSC, as well as in hiPSC. Reframed dysferlin showed a correct localization and maintained functional properties. Finally, re-framed mouse MuSC regenerated muscle, rescued dysferlin expression and localization, and repopulated the MuSC compartment when transplanted back into hEx44mut mice. Our study provides the first in vivo, proof-of-concept result for the efficacy of autologous cell replacement therapy in a muscular dystrophy using CRISPR-edited MuSC. Furthermore, we introduce an ideal model to investigate in vivo mRNA-mediated gene editing to treat LGMD2B by intervening directly on the relevant human sequence.

## Results

### MuSC from two patients with a founder *DYSF* frameshift mutation

We isolated primary MuSC populations from two related LGMD2B patients harboring a homozygous founder mutation (c.4872_4876delinsCCCC) in *DYSF* exon 44[27]. The mutation features a G nucleotide deletion in position c.4872 and a G > C nucleotide exchange in position c.4876 (Fig. 1a, b). The G deletion causes a frameshift and introduces a premature stop codon in exon 45, resulting in a complete absence of dysferlin protein. Patient-derived MuSC were >99% positive for the pan-myogenic marker desmin (DES), and expressed the myogenic stem and progenitor cell markers PAX7, MYOD, and MYF5, as well as the proliferation marker KI-67. We also obtained primary MuSC colonies from control donors (Supplementary Table 1).

### Screen for gRNAs to efficiently induce *DYSF* re-framing

We next developed a strategy to efficiently repair the mutation in patient MuSC and in skeletal muscle in vivo. As NHEJ is efficient in post-mitotic muscle fibers, we reasoned that we could harness the indel bias of some gRNAs in combination with Cas9 to rescue the *DYSF* exon 44 reading frame (Fig. 2a). We designed gRNAs that specifically target the mutant sequence to avoid targeting the wild-type exon 44 allele in the case of compound-heterozygous patients. We selected three gRNAs with high specificity profiles using the web tool CRISPOR[28] (Fig. 2b and Supplementary Table 2) and first tested our approach in hiPSC lines generated from both patients (Supplementary Fig. 1). We transfected plasmids encoding the enhanced specificity eSpCas9(1.1) variant[29], the single-guide RNA (sgRNA), and a Venus fluorescence reporter into hiPSC from a patient and a control, and enriched for Venus+ cells by FACS-sorting (Fig. 2b). We then analyzed the editing efficiency and indel frequencies by Sanger and next-generation amplicon sequencing (NGS) (Fig. 2c, d). No editing was detected with sgRNA#1. sgRNA#2 and #3 were specific to the mutant allele, as they showed editing activity only in patient hiPSC. Furthermore, sgRNA#3 led to a high rate of +1A insertions at the expected DSB site, which would rescue the exon 44 reading frame. We then established single cell-derived hiPSC colonies from FACS-sorted patient hiPSC transfected with eSpCas9(1.1) + sgRNA#3 and analyzed the genotype around the mutation. Most of the colonies (18/22) exhibited either heterozygous or homozygous editing. We found that 71% of all edited (20/28) alleles contained the +1A insertion at the expected DSB site (Fig. 2e, f). We next examined dysferlin expression in hiPSC and found it to be present at low levels in control hiPSC regardless of cell type of origin, and absent in patient hiPSC (Supplementary Fig. 2a). Patient-derived hiPSC clones carrying a homozygous +1A insertion showed a rescue of dysferlin protein (Fig. 2g and Supplementary Fig. 2b). Wild-type SpCas9 led to similar editing outcomes (Supplementary Fig. 3). Homology directed repair (HDR) of the c.4872_4876delinsCCCC mutation to wild-type could be achieved in patient-derived hiPSC, albeit with much lower efficiency than reframing (Supplementary Fig. 4).

### SpCas9 mRNA efficiently restores the *DYSF* exon 44 reading frame in MuSC

We previously established mRNA as a well-suited substrate to efficiently and safely deliver gene editing tools to human primary MuSC[25]. We now asked whether the same approach could induce efficient reframing in patient-derived primary MuSC. We transfected wild-type SpCas9 mRNA plus sgRNA#3 into MuSC from both patients as well as two control donors (Fig. 3a) and found >60% +1A insertion rates with an overall editing efficiency of >90%, as determined by NGS. As observed in patient hiPSC, the second most frequent indel was a −4N deletion (Fig. 3b–d). Moreover, the repair outcomes were consistent across all replicates from both patients, and the allele frequencies were maintained across cultivation time (Fig. 3c, d and Supplementary Fig. 5). No editing was detectable in MuSC from controls (Fig. 3c). We next performed GUIDE-seq[30] to identify potential off-target sites (OTS) for sgRNA#3. We identified 7 OTS that were also predicted by the in silico tool CrisprGold[31] (Fig. 3e, Supplementary Table 3). In addition, out of 81 OTS predicted by CRISPOR[28] (excluding the wild-type *DYSF* exon 44), we selected the 10 sites with the highest off-target score (three of which overlapped with the GUIDE-seq-identified OTS), plus all predicted exonic sites

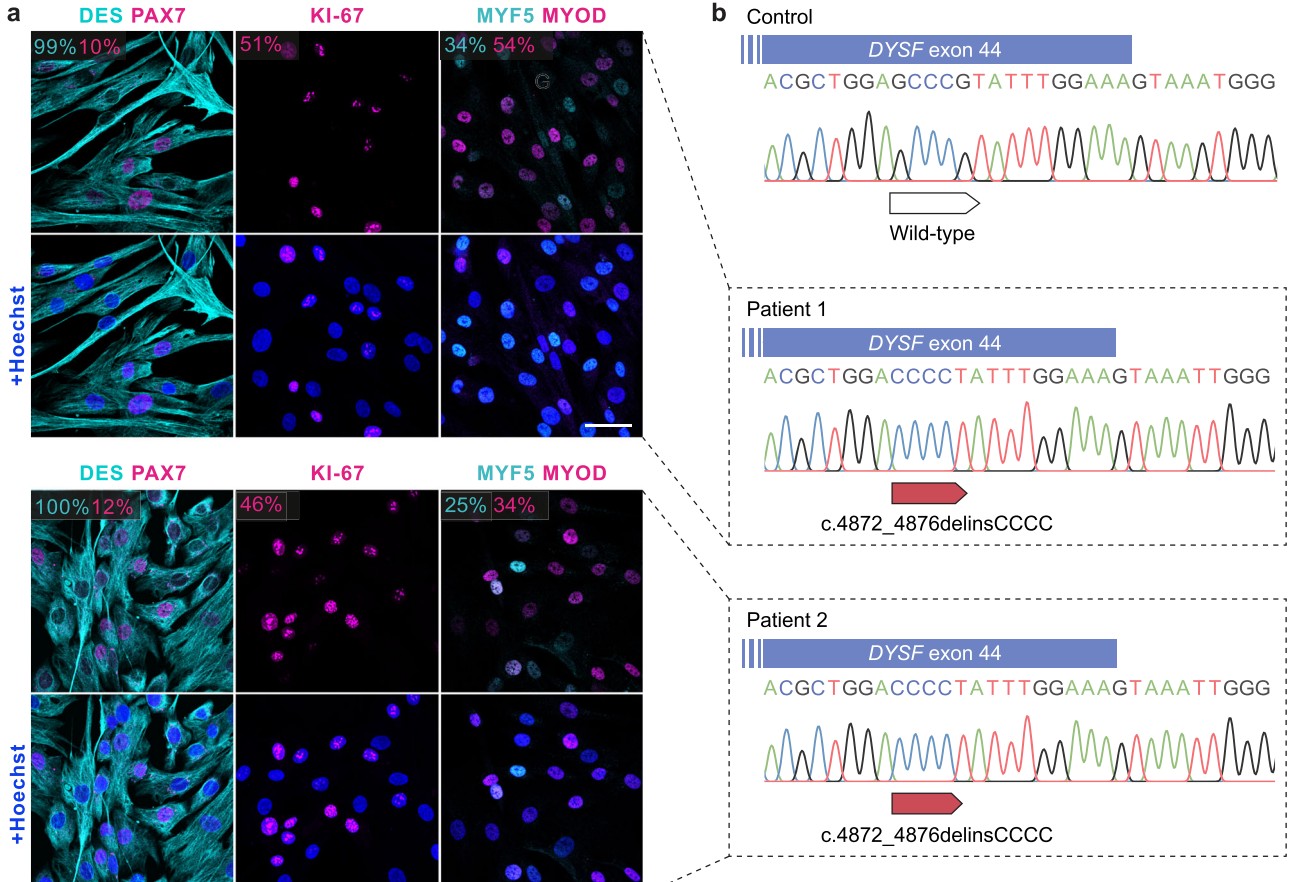

**Fig. 1 | Isolation of pure MuSC populations from two patients carrying a homozygous *DYSF* c.4872_4876delinsCCCC founder mutation. a** MuSC derived from patient 1 (upper panel) and 2 (lower panel) stained for the myogenic markers DES, PAX7, MYOD, MYF5, and the proliferation marker KI-67. Nuclei are stained with Hoechst. The percentage of cells expressing each marker is shown in the corresponding colors. Scale bars: 50 μm. Several MuSC populations (Patient 1: *n* = 2; Patient 2: *n* = 8) were prepared from a single muscle biopsy obtained from each patient. **b** PCR genotyping of *DYSF* exon 44 in genomic DNA from a control and the two patients carrying the homozygous c.4872_4876delinsCCCC mutation.

(Supplementary Table 3). We analyzed all these 21 sites by NGS in edited Vs unedited patient MuSC and detected a very low percentage (<0.3%) of modified reads likely corresponding to off-target activity at OTS 1–6 in edited cells, while no activity was detected at any of the other sites (Fig. 3f).

**Reframing rescues dysferlin protein expression and localization**
We performed Western blot analysis to assess whether restoring the *DYSF* exon 44 reading frame would rescue dysferlin protein expression in muscle cells. Edited patient MuSC showed a rescue of full-length dysferlin protein corresponding to ~60% of control levels, consistent with the +1A insertion rates. As expected, dysferlin was undetectable in unedited patient MuSC (Fig. 4a, b). We then induced terminal differentiation of MuSC to assess the localization of reframed dysferlin and identified sarcolemmal as well as cytoplasmic/vesicular dysferlin localization, indistinguishable from normal controls (Fig. 4c). Editing did not affect myogenic or proliferative properties of MuSC (Fig. 4d, e and Supplementary Fig. 6).

**Reframed dysferlin is functional**
The +1A nucleotide insertion induces a rescue of the *DYSF* reading frame but results in an exchange of four amino acids around the site of the mutation (T1622N, L1623A, E1624G and V1626L (Fig. 5a). No experimentally resolved 3D structure is available for full-length dysferlin or the C2F domain, where the c.4872_4876delinsCCCC mutation is located. We took several approaches to test for functional impact.

First, we superimposed the AlphaFold2-predicted 3D structure of dysferlin's C2F domain[32] with the experimentally resolved C2B domain of synaptotagmin-1[33,34]. We found that the $Ca^{2+}$-binding sites were highly conserved (Fig. 5b). The C2F mutations lead to only minor, conservative alterations of surface-exposed residues in a peripheral loop, which are unlikely to affect protein folding or $Ca^{2+}$-binding.

To experimentally address possible effects of the mutations, we purified recombinantly expressed wild-type (wt) and reframed human dysferlin C2F domains (C2Fwt and C2Fref, respectively) as fusions with the maltose binding protein (MBP), an affinity tag that increases solubility (Supplementary Fig. 7a). In thermal shift assays, both fusion proteins displayed a similar unfolding kinetics, indicating that the mutation does not majorly alter protein stability (Supplementary Fig. 7b, c). While the MBP moiety had a melting temperature ($T_M$) of ~52 °C (Supplementary Fig. 7c), the C2F domain and its variant appeared to unfold already at low temperatures without a clear melting point, which may hint at a low stability of the isolated domain. However, in Isothermal Titration Calorimetry (ITC) experiments, both the wt and reframed C2F domain exhibited similar high affinities for $Ca^{2+}$ ions, with dissociation constants ($K_d$) of (70 ± 30) nM and (120 ± 60) nM, respectively (Supplementary Fig. 7d). Thus, the mutations do not appear to majorly affect the $Ca^{2+}$-binding function of the C2F domain.

Finally, we assessed the capability of reframed dysferlin to engage in sarcolemmal repair. We injured either edited and/or unedited patient myotubes with a laser and determined the localization of dysferlin at the repair site[35]. The site is characterized by accumulation

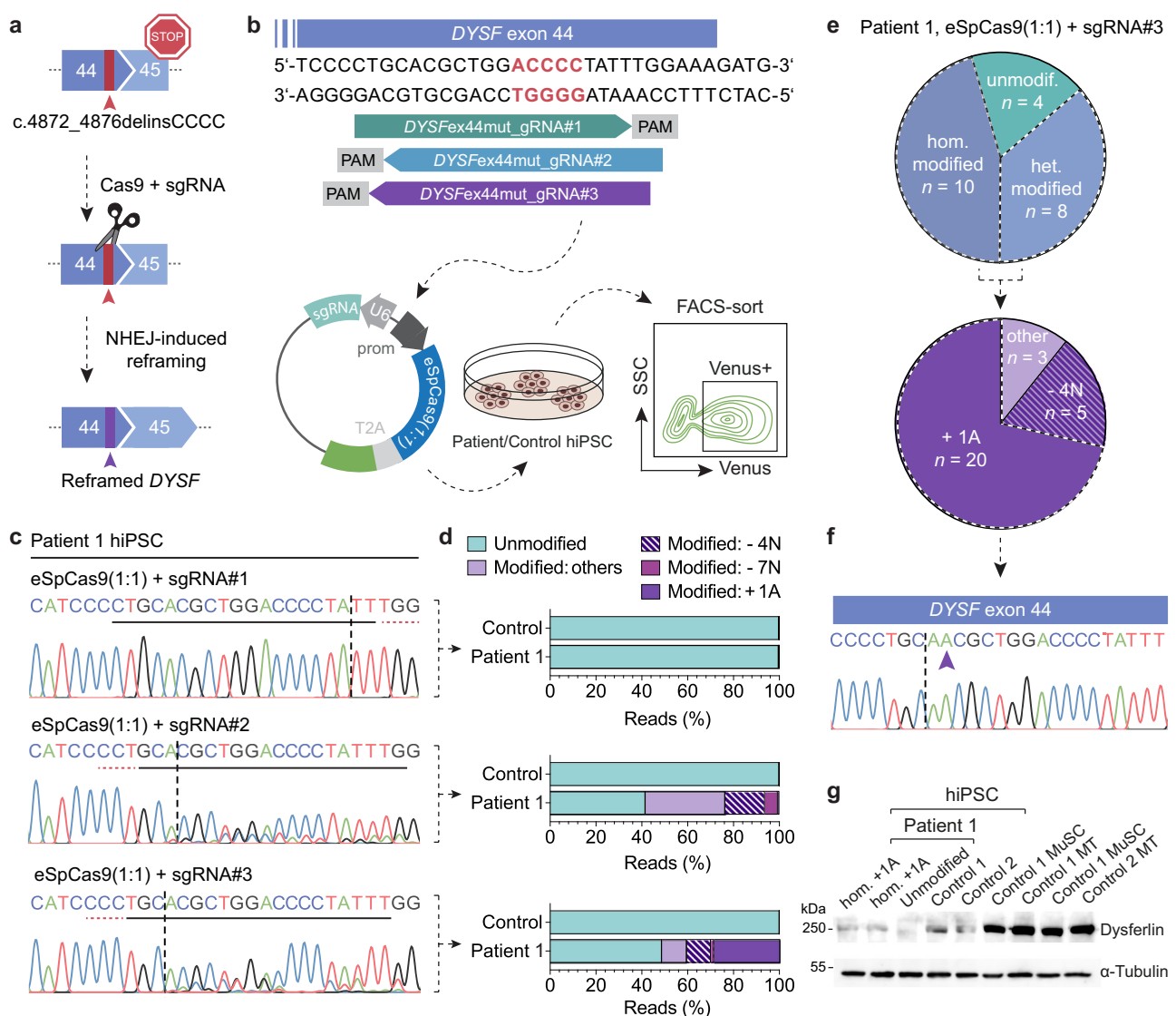

**Fig. 2 | Screening for mutation-specific sgRNAs to rescue the *DYSF* exon 44 reading frame in patient hiPSC. a** Strategy to rescue the *DYSF* reading frame by Cas9-induced NHEJ. **b** hiPSC from patient 1 and a control (not carrying the *DYSF* exon 44 mutation) were transfected with a plasmid encoding eSpCas9(1:1) in combination with three different mutation-specific sgRNAs. Venus positive cells were selected by FACS-sorting and processed for DNA analysis via NGS. **c** Sanger-sequencing chromatograms of patient 1 hiPSC after FACS-sorting. The protospacer sequences are underlined in black. The PAM sequences are underlined in red. The dotted vertical lines indicate the expected DSB sites. **d** Allele frequencies of Venus+ patient 1 and control hiPSC transfected with eSpCas9(1.1) and sgRNA#1-3 (as on the left) were determined by NGS. The most frequent indels are indicated. **e** Single cell-

derived clones were expanded from Venus+ patient 1 hiPSC transfected with eSpCas9(1.1) and sgRNA#3 (*n* = 22 clones). Upper pie chart: Distribution of unedited, heterozygously edited and homozygously edited clones. Lower pie chart: Indel distribution among all edited clones. **f** Sanger-sequencing chromatogram from an hiPSC clone carrying a homozygous +1A insertion at the DSB site (purple arrow). **g** Western blot analysis of dysferlin protein expression in two edited hiPSC clones from patient 1, homozygous for the +1A insertion, compared to unedited hiPSC from patient 1 and two controls, as well as MuSC and myotubes (MT) from two controls. α-tubulin was used as loading control (*n* = 1 independent experiment). Source data are provided as a Source Data file.

of annexin A1. Reframed dysferlin accumulated at the repair site following membrane wounding (Fig. 5c).

## Generation of a novel mouse model with a humanized *DYSF* exon 44 carrying the founder mutation

To investigate the therapeutic potential of our gene editing approach in vivo, we generated a mouse model carrying the human exon 44, with and without the c.4872_4876delinsCCCC mutation, in the endogenous *Dysf* locus (Fig. 6a and Supplementary Fig. 8). Both the humanized wild-type and mutant exon 44 were normally spliced in mouse muscle (Supplementary Fig. 9). Mice homozygous for the wild-type allele (*Dysf*<sup>hEx44wt/hEx44wt</sup>) showed normal levels of dysferlin mRNA and protein, a normal dysferlin localization pattern, and no

pathological phenotype (Fig. 6b–e, Supplementary Figs. 10–12). In contrast, mice homozygous for the exon 44 mutation (*Dysf*<sup>hEx44mut/hEx44mut</sup>) had strongly reduced dysferlin mRNA levels and completely lacked dysferlin protein (Fig. 6b–d). Myopathologically, muscular dystrophy was evident by central nuclei, split fibers, necrotic fibers and immune infiltrates with a disease onset at around 8 weeks of age (Fig. 6e and Supplementary Fig. 10a, b). Changes in fiber size distribution were only moderate in TA and not apparent in Soleus muscles at 20 weeks, worsening slightly at 40 weeks, whereas they were very dramatic in quadriceps (Supplementary Fig. 10c, d). Fibrosis was very severe in quadriceps muscles at 40 weeks (Supplementary Fig. 11), where eMyHC+ regenerating fibers were found in small clusters and macrophage infiltration was very widespread (Supplementary Fig. 12). CD3+

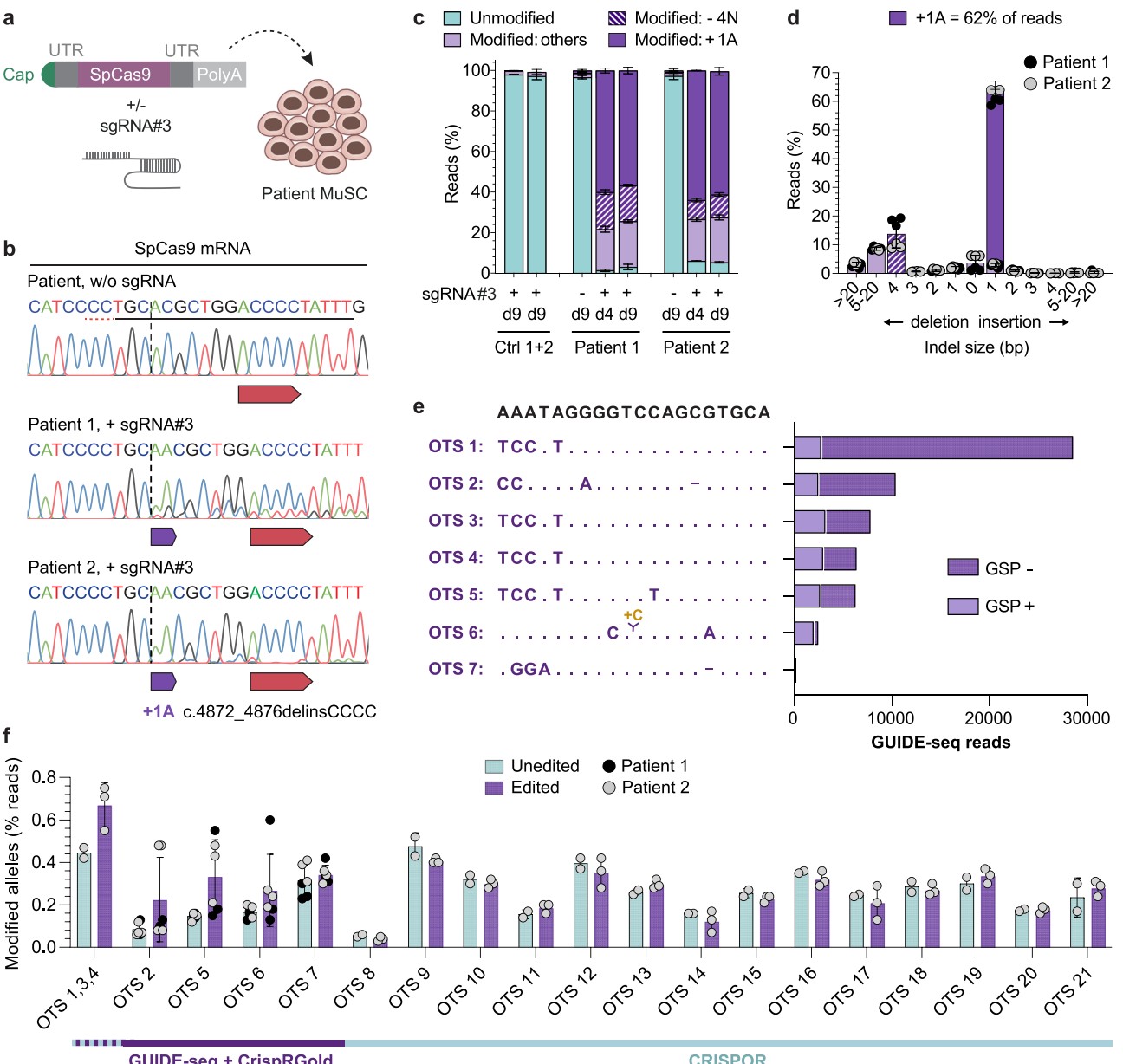

**Fig. 3 | SpCas9 mRNA reframes *DYSF* exon 44 with >60% efficiency in MuSC from both patients with high precision and safety. a** Schematic overview of experimental workflow. Patient and control MuSC were transfected with SpCas9 mRNA (3 μg per 150,000 cells), with or without sgRNA#3 (2 μg per 150,000 cells). **b** Sanger-sequencing chromatograms around the target site from patient MuSC transfected with or without sgRNA#3. The protospacer and PAM sequences are underlined in the upper unedited chromatogram. The dotted vertical line indicates the expected DSB site. **c** Allele frequencies in control and patient MuSC transfected with SpCas9 mRNA, with or without sgRNA#3, at day 4 and 9 post transfection determined by NGS (*n* = 3 replicates per donor and time point, mean ± SD). Data from the two control MuSC populations are plotted together. Ctrl: Control.

**d** Frequency distribution of all indels in MuSC from the two patients at day 4 post-transfection with SpCas9 mRNA and sgRNA#3 (*n* = 3 replicates per patient, mean ± SD). **e** OTS identified by GUIDE-seq. The OTS sequences are aligned to the on-target site (top). The total number of GUIDE-seq reads aligned to each OTS from all samples, using both the GSP+ and GSP- primers, is plotted on the right. **f** All OTS identified by GUIDE-seq/CrispRGold (OTS 1-7), plus the top 10 and all exonic OTS predicted by CRISPOR (OTS 1, 3, 4 and OTS 8-21) for sgRNA#3 (Supplementary Table 3) were analyzed by NGS in patient MuSC collected 4–9 days after transfection with SpCas9 mRNA, with (*n* = 3) or without (*n* = 3 for OTS 2, 5, 6 and 7; *n* = 2 for all other OTS) sgRNA#3 (mean ± SD). Source data are provided as a Source Data file.

T-lymphocytes were found in small infiltrates, whilst CD8+ and CD20+ T- and B-lymphocytes, respectively, were scarce and occurred isolated (Supplementary Fig. 13).

## Highly conserved gene editing outcomes restore dysferlin expression in MuSC from hEx44mut mice

Having established the humanized mice carrying the founder mutation, we next tested the reframing strategy in the respective mouse MuSC (Fig. 7a and Supplementary Fig. 14). We used exactly the same sgRNA and

mRNA as for the human MuSC and observed >90% editing in mouse MuSC, with ~60% +1A nucleotide insertions at the DSB site (Fig. 7b, c). The mRNA quantification, protein analysis by Western-blot and immunos-taining revealed a robust dysferlin rescue as in patient cells (Fig. 7d–g).

## Exon 44 re-framed murine MuSC regenerate muscle and rescue dysferlin in hEx44mut mice

Finally, we investigated the regenerative potential of reframed MuSC in an autologous cell replacement therapy setting. We

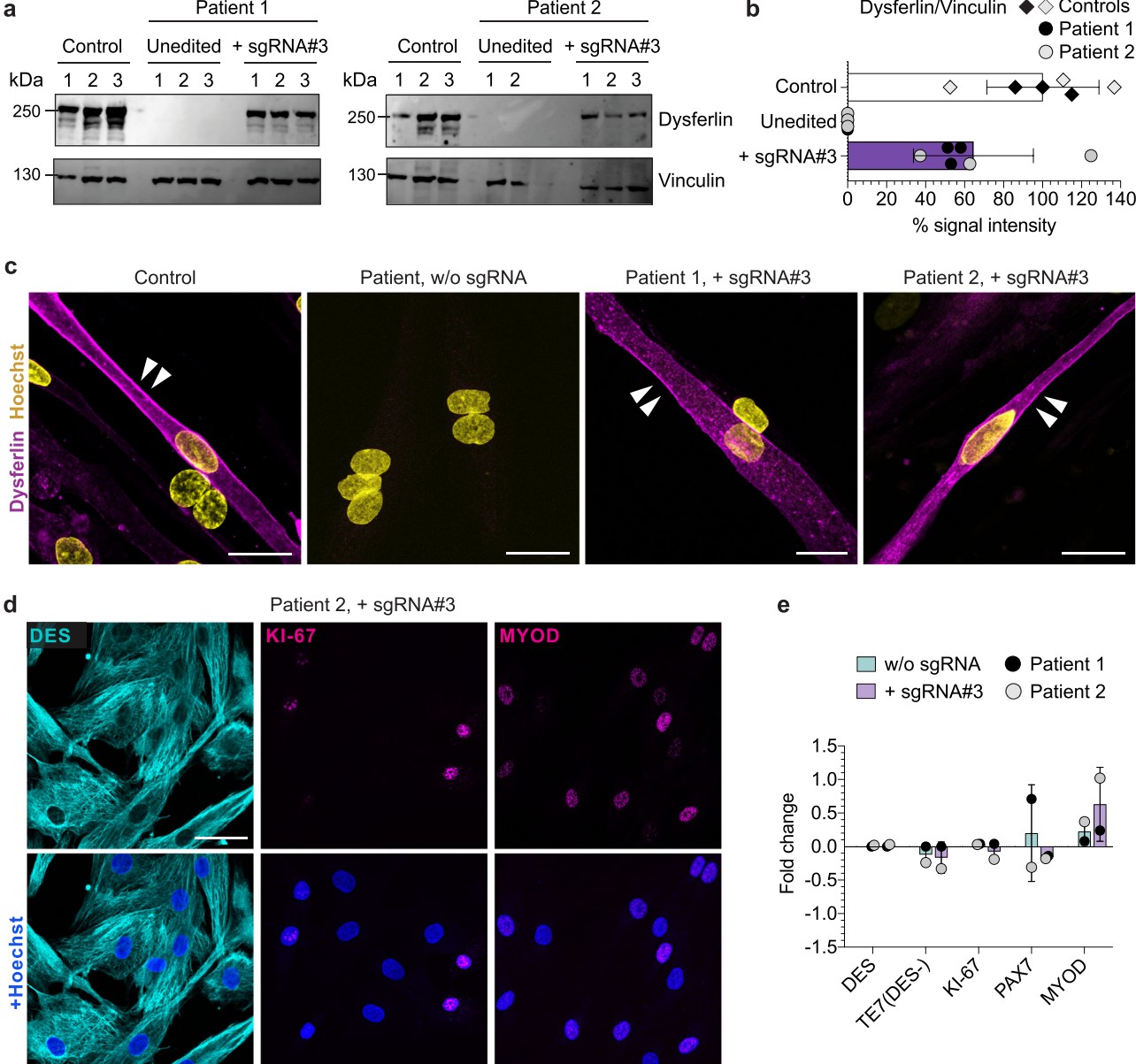

**Fig. 4 | SpCas9 mRNA-mediated reframing efficiently rescues dysferlin protein expression and correct localization in patient MuSC while maintaining their myogenic profile. a** Western blot analysis of dysferlin protein expression in edited patient MuSC. Unedited patient and control MuSC were used as controls. Vinculin was used as loading control. **b** The intensity of the dysferlin bands from **a** was quantified using ImageJ and normalized to Vinculin ($n = 3$ replicates, mean ± SD). **c** Dysferlin immunostaining of terminally differentiated control and patient MuSC transfected with SpCas9 mRNA, with or without sgRNA#3. Dysferlin membrane localization is indicated by white arrows. Scale bars: 25 μm. **d** Edited patient MuSC stained for DES, PAX7 and MYOD. Scale bar: 50 μm. **e** Fold-change of the percentage of patient cells expressing myogenic and proliferation markers at day 5 after transfection of SpCas9 mRNA, with or without sgRNA (normalized to untransfected cells) ($n = 1$ independent experiment per patient, mean ± SD). Source data are provided as a Source Data file.

transplanted exon 44 re-framed primary MuSC (Supplementary Fig. 15) into *tibialis anterior* (TA) muscles of early symptomatic (≈12 weeks old) and late symptomatic (75 weeks old) homozygous hEx44mut mice. We tested three regimens of pretreatment of recipient muscles prior to transplantation: (1) no pretreatment, (2) irradiation, and (3) cardiotoxin plus irradiation (Fig. 8a). Cardiotoxin is derived from snake venom and triggers muscle regeneration by inducing an acute muscle injury. We found dysferlin-positive donor-derived myofibers in all groups, albeit with different efficiencies. As expected, no pretreatment resulted in the least dysferlin positive fibers (Fig. 8b, c). In CTX-injected and irradiated-only muscles, we found numerous donor-derived myofibers with a strong membrane as well as a cytoplasmic reticular

dysferlin localization pattern (Fig. 8b, d, e and Supplementary Fig. 16). In addition, we found Pax7-expressing cells adjacent to donor-derived, dysferlin-expressing myofibers in both CTX-injected and irradiated-only muscles (Fig. 8f, g). Moreover, we also found that some donor-derived cells had repopulated the MuSC niche in the graft, as shown by their sublaminal localization and expression of Pax7 (Fig. 8h). We further investigated the presence of a cellular immune response against donor cells in successfully grafted non-pre-injured late symptomatic host muscle. CD3+ and CD8+ cell infiltrates were present in both the grafted and the contralateral TA muscles, without an apparent enrichment in the grafted muscle (Supplementary Fig. 17). CD20+ cells were not detected.

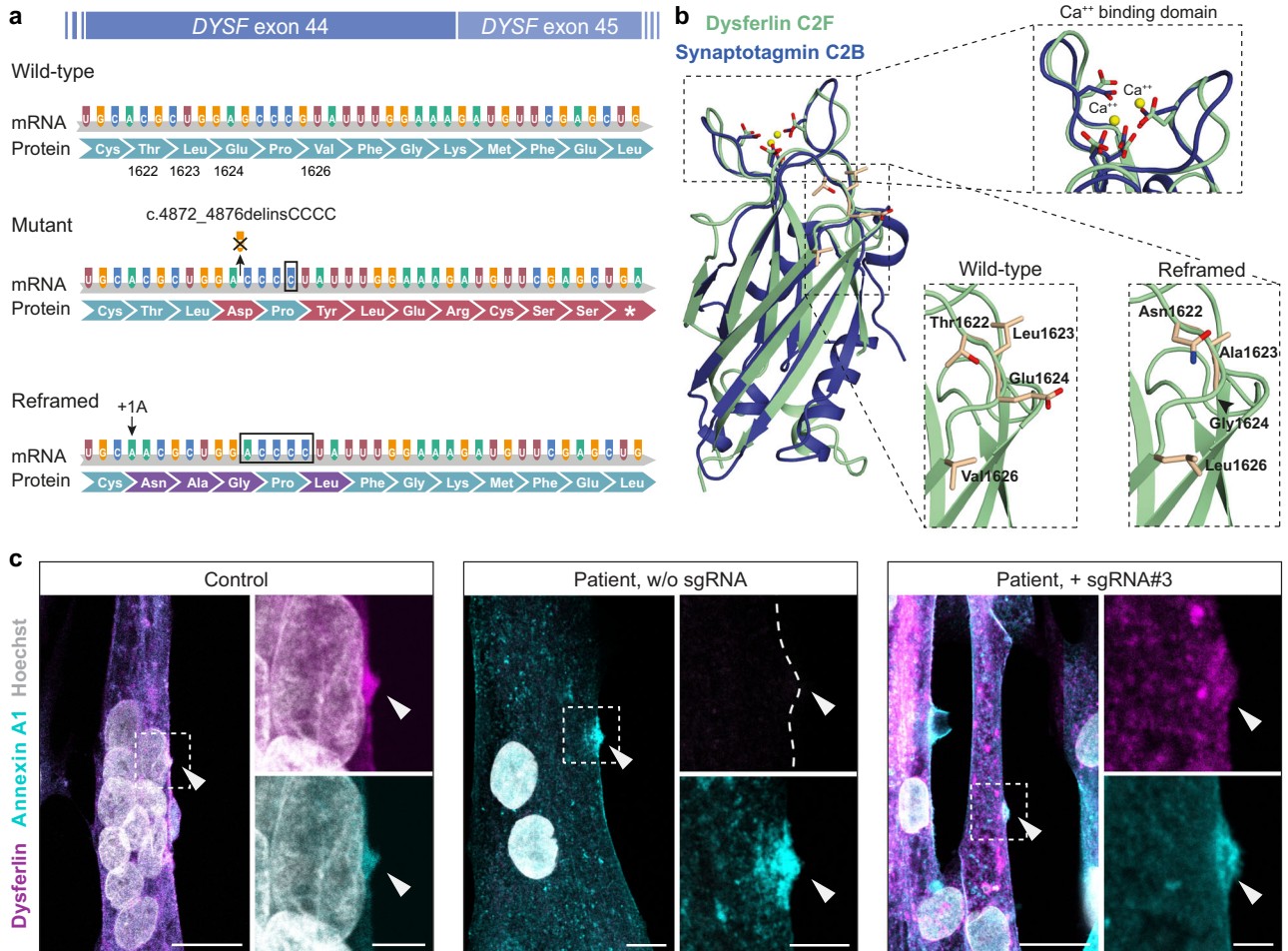

**Fig. 5 | Reframed dysferlin maintains functional properties. a** Scheme showing the mRNA and amino-acid sequence of wild-type, mutant, and reframed *DYSF* exon 44/45. The c.4872_4876delinsCCCC founder mutation introduces a frameshift in exon 44 that results in a premature stop codon in exon 45. The SpCas9-induced +1A insertion rescues the reading frame and results in an exchange of four amino acids (p.1622, p.1623, p.1624, and p.1626) compared to the wild-type protein. **b** Upper: Superposition of the experimentally determined Ca$^{2+}$-bound C2B domain of rat synaptotagmin-1 (pdb 1TJX) with the AlphaFold2-predicted C2F domain of human dysferlin (Expasy accession number: O75923). The Ca$^{2+}$ binding site is highly conserved in both domains (magnified top view). Lower: The magnification shows the site of the four mutated amino acids in a surface-exposed loop connecting the β-sandwich with the Ca$^{2+}$-binding site of the C2F domain. **c** Dysferlin and annexin A1 immunostaining of unedited and edited patient myotubes, compared to control myotubes, performed up to 20 min after laser irradiation. Arrows indicate the wounding area (repair site). Nuclei are counterstained with Hoechst. Scale bars: 20 μm (lower magnification) and 5 μm (zoomed-in images). The experiment was conducted 3 independent times with similar results, and 13–26 myotubes were successfully injured and analyzed per condition (Control: $n = 13$; Patient, unedited: $n = 17$; Patient, edited: $n = 26$).

## Discussion

We demonstrate a highly predictable and consistent Cas9-induced reframing that leads to a rescue of full-length dysferlin on mRNA, protein, and functional levels in the context of an LGMD2B-disease-causing founder *DYSF* mutation. Our evidence is provided in hiPSC, primary human MuSC, and in MuSC from a newly generated humanized mouse model. Transplantation of gene-repaired primary MuSC in an autologous context resulted in muscle regeneration and rescue of dysferlin protein in vivo by the donor cells without the need of immune modulation. Donor-derived cells also repopulated the MuSC niche, indicating that they could provide a long-term contribution to healthy muscle homeostasis and regeneration. These results set the stage for the clinical application of a cell replacement therapy using repaired, autologous MuSC in patients, as well as the development of an effective in vivo gene-editing strategy for this as yet untreatable disease.

The editing outcome of some sgRNAs in combination with Cas9 is biased towards a certain indel signature[36]. While deletions are mainly associated with microhomology regions, driving repair through alternative end joining pathways, insertions are dominated by the staggered-end cleavage mechanism. This indel bias can be predicted to some extent and has been harnessed to therapeutically edit pathogenic DNA variants[37,38]. The +1A indel observed here can be explained by staggered-end cleavage and comprises a constant rate of Cas9-induced editing events. Moreover, we found that the frequency of the +1A insertion is reproducible across donors, cell-types, and even across species, suggesting a highly conserved cleavage pattern and DNA repair pathway choice at this site.

Exon reframing has been observed in skeletal and cardiac muscle in the context of NHEJ strategies to disrupt splice sites for *DMD* exon skipping using several Cas nucleases[39–42]. Clinical evidence for the success of exon skipping strategies is still scarce. Preclinically, we could previously demonstrate successful *DYSF* exon 38 skipping to circumvent a missense mutation[43]. However, *DYSF*, lacking repetitive sequences, may be an even poorer candidate for exon-skipping approaches than *DMD*, and strategies to rescue a full-length functional protein are more relevant.

To investigate the therapeutic relevance of systemic CRISPR machinery administration, it is thus highly advantageous to have mouse

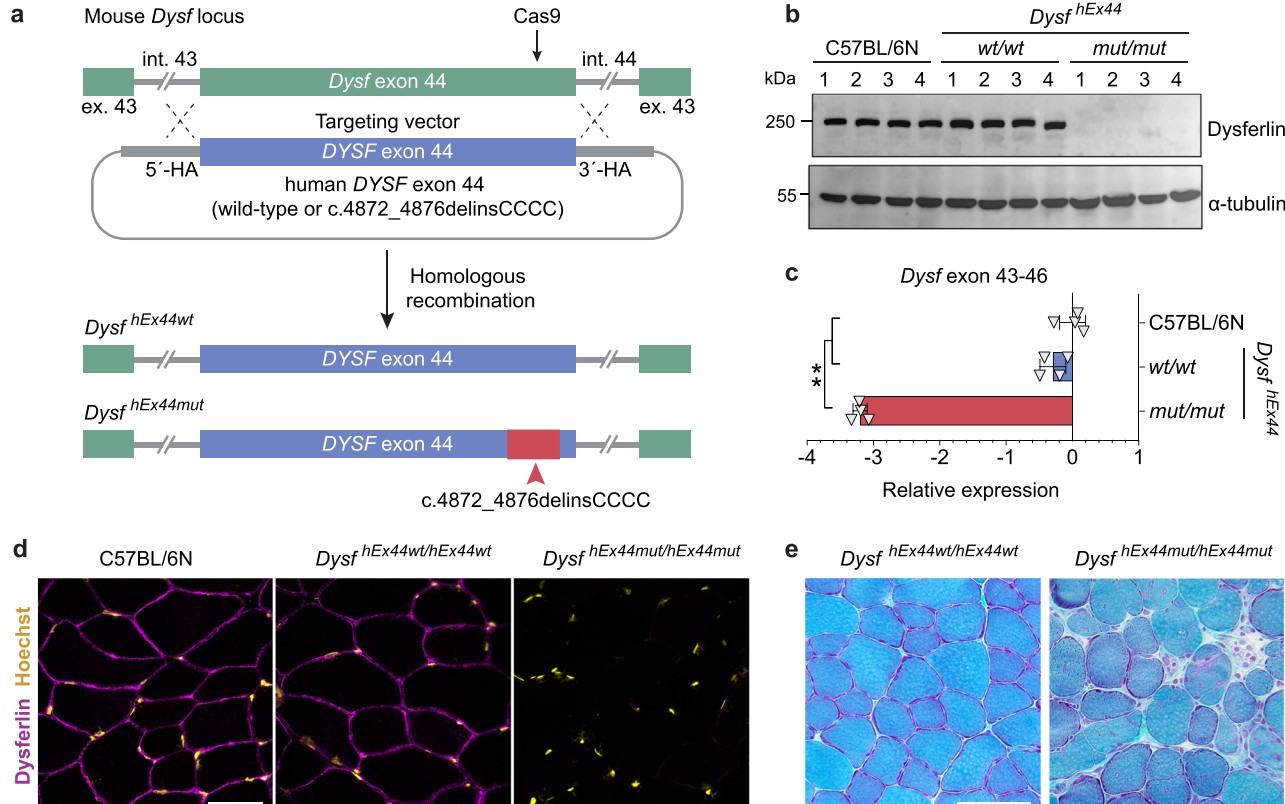

**Fig. 6 | Generation and characterization of a novel humanized LGMD2B mouse model carrying the c.4872_4876delinsCCCC founder mutation. a** Schematic overview of the strategy to generate the transgenic mice. SpCas9 and a gRNA were used to induce a DSB at the 3′ end of the murine *Dysf* exon 44 in C57BL/6N mouse zygotes. A targeting vector containing either the human wild-type or mutant *DYSF* exon 44 flanked by 1.4 kb homology arms (HA) was provided to enable exchange of the murine exon 44 for the human exon 44 via homologous recombination. The resulting knock-in alleles were called hEx44wt (wild-type human exon 44) and hEx44mut (human exon 44 with the c.4872_4876delinsCCCC mutation). **b** Western blot analysis of dysferlin protein expression in quadriceps muscles of 12-week-old homozygous hEx44wt and hEx44mut male mice compared to age- and gender-matched wild-type C57BL/6N mice ($n = 4$ per genotype). **c** RT-qPCR analysis of dysferlin mRNA (relative to *Gapdh*) in quadriceps muscles from 12-week-old homozygous hEx44wt and hEx44mut male mice compared to age- and gender-matched wild-type C57BL/6 N mice ($n = 4$, mean ± SD). Kruskal-Wallis with post hoc Dunn's test ($p = 0.0065$). **d** Dysferlin immunostaining of *M. tib. ant.* from homozygous hEx44wt and hEx44mut mice. Nuclei were stained with Hoechst. Scale bar: 50 μm. **e** Gomori's Trichrome staining of *M. tib. ant.* from 40-week-old homozygous hEx44wt and hEx44mut mice. Scale bar: 100 μm. Source data are provided as a Source Data file.

models that carry the mutation in the human sequence context. While this state-of-affairs is not realistic for all mutations, it is justified for founder mutations with a high carrier frequency, where many patients could benefit from a specific editing approach. Cas9-induced re-framing was recently shown in a *DMD* mouse model carrying a humanized *DMD* exon 51, but the molecular signatures of Cas9-induced edits in mouse versus human cells were not investigated in detail[44].

Many patients with LGMD2B have been diagnosed carrying homozygous or compound heterozygous missense mutations distributed throughout the entire length of the gene. In mutations like *DYSF* p.Leu1341Pro and p.Val67Asp, the effects on protein folding and function have been demonstrated and are likely sufficient to cause progressive muscle wasting[45,46]. The effect on protein structure of other *DYSF* missense mutations is not as obvious. The possibility that splicing alterations rather than single amino acid exchanges are causing the severe disease course has been demonstrated for *DYSF*[47] and *SGCA*, where an LGMD2D/R3-causing mutation that was previously interpreted as a missense variant, instead induces aberrant splicing and completely abolishes the expression of full-length *SGCA* mRNA[24]. Exonic splicing signals appear to be undisturbed by our reframing approach as both mRNA and protein were rescued to levels similar to the +1A insertion rates.

In reframed dysferlin, as generated here, four amino acids differ from the wild-type sequence. We carefully studied the effect of these amino acid substitutions on protein expression, localization, structure

and function. We showed that (1) dysferlin protein abundance is similar in wild-type and edited human and mouse muscle cells, arguing against increased degradation, (2) the protein localization pattern is indistinguishable from wild-type and (3) the protein accumulates at the repair site in response to membrane injury. Moreover, based on structural considerations, the ensuing four mutations in the reframed variant do not affect protein folding or the $Ca^{2+}$-binding function of the dysferlin C2F domain, as also confirmed in biophysical experiments. Accordingly, in all tested features, reframed dysferlin behaves basically identical to the wild-type protein.

Several protocols have been developed to generate pluripotent stem cell-derived myogenic cells[48–50], making them a promising source for cell replacement therapies for muscular dystrophy[21,51], although issues regarding their purity and maturity are yet to be solved[52]. In our study, we used hiPSC to establish gene editing strategies for muscular dystrophy because they are available in unlimited numbers. In these circumstances, our readouts are usually restricted to analyzing editing at the genomic level because muscle proteins are not expressed in undifferentiated hiPSC. We detected dysferlin in all examined control hiPSC lines. Its expression is unlikely to be a remnant of the cell type of origin, as we found it in hiPSC derived from MuSC, PBMC or skin fibroblasts, and also in later passages post-reprogramming. Remarkable is the fact that dysferlin has a different expression pattern from other muscle proteins associated with muscular dystrophy. The finding that dysferlin

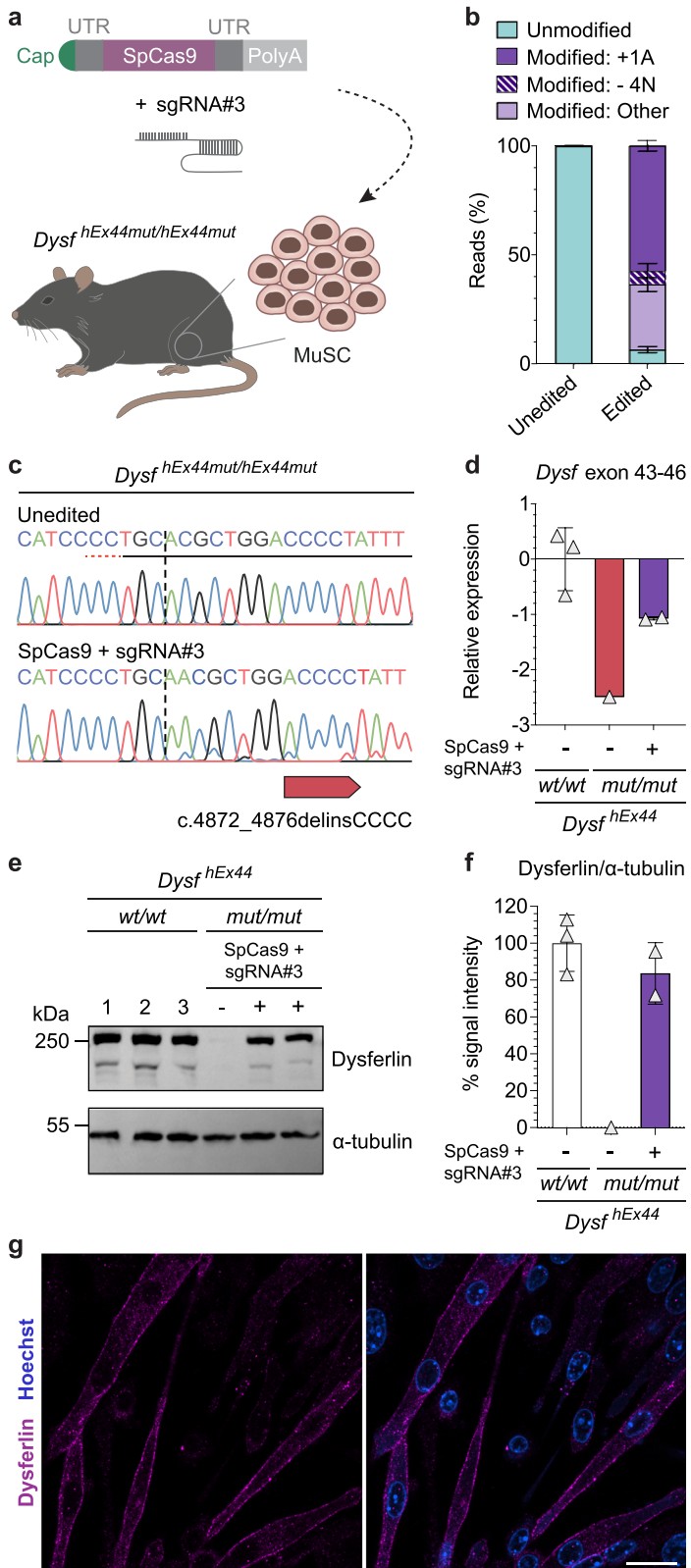

**a**

**b**

Unmodified
Modified: +1A
Modified: - 4N
Modified: Other

**c**

*Dysf* hEx44mut/hEx44mut

Unedited
CATCCCCTGCACGCTGGACCCCTATTT

SpCas9 + sgRNA#3
CATCCCCTGCAACGCTGGACCCCTATT

c.4872_4876delinsCCCC

**d**

*Dysf* exon 43-46

**e**

*Dysf* hEx44

wt/wt   mut/mut

SpCas9 + sgRNA#3

**f**

Dysferlin/α-tubulin

**g**

is consistently expressed in hiPSC, obviously facilitates the evaluation of editing outcomes on protein levels without lengthy differentiation protocols. However, these results also point towards a role different from membrane repair. Dysferlin functions are numerous and speculative, including involvement in metabolic and signaling pathways[9,10,23]. However, its roles beyond sarcolemmal repair remain yet poorly understood and our findings raise interesting questions.

Clinically, dysferlinopathy differs from other LGMDs in the almost equal involvement of proximal and distal muscles. Particularly discomforting for daily activities and independence in eating, drinking, and personal hygiene is the early involvement of the finger flexors. This small muscle group is an ideal candidate for a cell replacement therapy and deserves particular clinical attention, even if different in vivo gene replacement or gene repair strategies come into place. Skeletal muscle,

**Fig. 7 | SpCas9 mRNA reframes *DYSF* exon 44 and rescues dysferlin protein expression in MuSC from hEx44mut mice. a** Schematic overview of experiment setup. MuSC isolated from hind limbs of homozygous hEx44mut mice were transfected with SpCas9 mRNA plus sgRNA#3 or GFP mRNA. **b** Allele frequencies in unedited ($n = 3$) and edited ($n = 6$) MuSC from homozygous hEx44mut mice determined by NGS (mean ± SD). **c** Sanger-sequencing chromatogram of MuSC from homozygous hEx44mut mice transfected with SpCas9 mRNA plus sgRNA#3 compared to unedited cells. The protospacer and PAM sequences are underlined. The dotted vertical line indicates the expected DSB site. **d** Relative *Dysf* mRNA expression in edited and unedited MuSC from homozygous hEx44mut mice normalized to MuSC from homozygous hEx44wt mice (hEx44wt: $n = 3$; hEx44mut, unedited: $n = 1$, hEx44mut, edited: $n = 2$, mean ± SD). **e** Western blot analysis of dysferlin protein expression in edited and unedited MuSC from homozygous hEx44mut mice. MuSC from homozygous hEx44wt mice were used as control. **f** Quantification of dysferlin signal relative to α-tubulin from **e** using ImageJ. **g** Dysferlin immunostaining of reframed hEx44mut MuSC after differentiation into myotubes. Nuclei were stained with Hoechst. Scale bar: 20 μm. Source data are provided as a Source Data file.

being the largest organ in the body, is an accessible source of primary MuSC with a well-defined identity. They are applicable in an autologous fashion, provided that they can be efficiently edited to correct the muscular-dystrophy-causing genetic defect as shown here. We recently showed that gene edited patient MuSC can regenerate muscle and repopulate the stem cell compartment in xenografts, indicating that they could support healthy muscle homeostasis and regeneration long-term and thereby have a sustained therapeutic effect[24].

Incapacitation of resident MuSC, e.g., by radiation, and the presence of a regenerative environment in the host muscle enhance donor cell engraftment in mice[53,54]. Consistently, the largest number of donor-derived myofibers per graft was found when both irradiation and CTX injury were applied. Importantly, donor cells gave rise to bundles of myofibers histologically very similar to native muscle in terms of size and distribution. Transplantation outcomes in CTX-treated muscle were variable, likely due to technical limitations of the single cell injection and the fact that the effect of the CTX did not always reach the entirety of the muscle. In irradiated-only muscles, donor fibers were consistently found only along the injection trajectory, where muscle regeneration takes place, in accordance to previous studies[24,54,55]. Moreover, under physiological conditions in the natural course of disease progression, in the absence of experimentally induced injury, dysferlin positive fibers also engrafted. Whether or not pretreatment of muscle will be beneficial or even mandatory in clinical settings warrants further research. In addition to generating new fibers, donor cells gave rise to satellite cells that colonized the stem cell niche. Hence, in principle, these could support muscle turnover and regeneration long-term, and gradually replace the old myonuclei with new, gene-corrected nuclei. To maximize the size of the graft and therapeutic impact, high-density injections would likely be required.

The long-term crosstalk between the donor cells and the host muscle in terms of safety, restoration of muscle strength and potential immune responses against the neo-antigens expressed by the gene-edited cells, are important questions that require further in-depth analysis. In the more studied field of AAV gene supplementation therapy for DMD, if and under what conditions immune responses against dystrophin are a concern remains controversial[56–58]. Restoring endogenous dysferlin levels through an autologous muscle stem cell-based gene therapy is a different scenario. However, the possibility of eliciting an immune response against the reconstituted dysferlin cannot be ruled out and might be influenced by whether the patient has residual amounts or a complete absence of protein.

We suggest that in vivo gene editing will be an additional step towards systemic therapy and delivering Cas9 as mRNA eliminates risks of genomic integration and prolonged exposure, both of which can result in increased off-target mutagenesis and immunological problems[56]. Clinical trials of in vivo mRNA delivery of CRISPR enzymes for other diseases are underway and have reported encouraging results[59,60]. Targeting muscle systemically with non-viral vectors is still a challenge, but recent efforts to deliver Cas9 mRNA using lipid-nanoparticle carriers have shown promising results in DMD mice, and further developments are guaranteed[61]. Whether or not non-viral in vivo delivery vehicles will succeed in editing resident MuSC, which would be required for a life-long effect, is still unknown. In that context, autologous transplantation of gene repaired MuSC could be envisioned as an adjuvant treatment to ensure a long-term reservoir of gene-corrected stem cells.

Dysferlin has a very large coding sequence and at least 14 different transcript variants that are tissue-specifically regulated[62]. Therefore, repairing disease-causing mutations in situ has many advantages over gene supplementation strategies. Both for ex vivo and in vivo applications, achieving the highest possible rate of therapeutic edits is crucial for gene editing treatments. Restoring dysferlin protein levels as little as 10% may be sufficient to rescue the disease phenotype[63,64]. Prime editing could repair the sequence to wild-type[65] and recent developments in this technology highlight interesting prospects[66–68]. However, prime editing is yet to be reported in skeletal muscle in vivo and, although smaller variants have been engineered, the prime editor is a substantially larger protein for in vivo delivery with a viral component that raises some additional immunological concerns[67]. Cas9-induced reframing efficiencies of >60% with analogous protein rescue rates, as shown here, would likely be curative. Furthermore, the ratio of templated insertions versus other types of indels can be increased by targeting bacterial polymerases to the Cas9-induced DSB site, although this approach also increases the proportion of large deletions, rendering it unsafe for therapy in its current form[69].

Biosafety is a crucial aspect in the translation of gene editing therapies to the clinical setting. We applied a comprehensive approach for off-target nomination based on both in silico prediction algorithms and GUIDE-seq, and found very low frequency of off-target editing (indel) events at six of the nominated sites by targeted amplicon sequencing. These sites are intronic or intergenic, and indels are unlikely to disturb gene expression. However, all off-target events need to be cautiously regarded. In addition to smaller indels, Cas9-induced DSBs at both on- and off-target sites can result in large structural variants and aneuploidy, which can persist over cell divisions[70–72]. How often those outcomes occur in muscle cells and how permissive these are to maintaining such chromosomal abnormalities warrants further studies. We show that the allele frequencies resulting from different DNA repair outcomes at the target site remain constant in edited MuSC from both patients between day 4 and day 8/9 post-editing. This strongly suggests an absence of clonal drift due to proliferative advantage or disadvantage of edited MuSC subpopulations during the observation period. Long-term monitoring of the clonality of ex vivo expanded and in vivo engrafting gene-edited MuSC populations is an interesting future prospect. For clinical translation, sensitive methods to detect structural variants, as well as biodistribution and tumorigenicity studies, will be required. Ultimately, provided that robust pipelines are in place to uncover potential pitfalls, the risk-benefit ratio will need to be evaluated for each indication.

In sum, we developed a toolbox across cell-types and species to translate repair of an important *DYSF* founder mutation into clinical application and demonstrate the first in vivo rescue of a muscular dystrophy by autologous transplantation of gene-edited primary MuSC. We suggest that our findings could further advance novel therapies for muscular diseases.

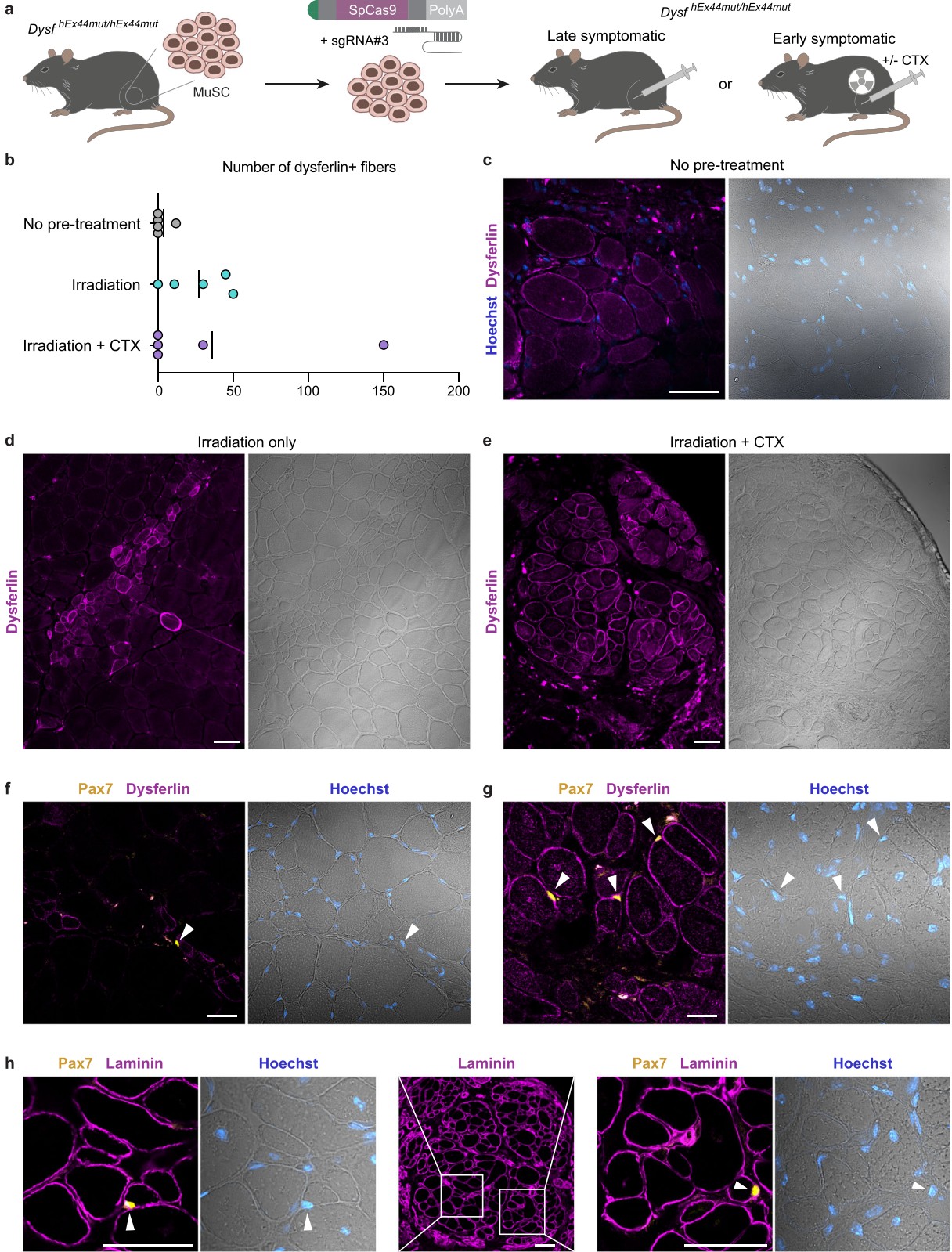

## Methods

### Study approval

The regulatory agencies (EA2/051/10 and EA2/175/17, Charité−Universitätsmedizin Berlin) approved the studies and written informed consent, including permission to generate hiPSC lines, was obtained from donors or legal guardians. Animal experiments were performed under the license numbers G0162/12, G0111/17, G0301/18, and G0223/20 (Landesamt für Gesundheit und Soziales−LaGeSo−Berlin, Germany).

### Patients and control donors

All donors were seen at the Outpatient Clinic for Muscle Disorders of the Charité−Universitätsmedizin Berlin (Supplementary Table 1).

**Fig. 8 | Exon 44 re-framed murine MuSC regenerate muscle and rescue dysferlin in hEx44mut mice. a** MuSC were isolated from homozygous hEx44mut donors, transfected with SpCas9 mRNA and sgRNA#3, and transplanted into the TA muscles of late (75 weeks old) or early (11–14 weeks old) symptomatic homozygous hEx44mut recipients. Prior to grafting, late symptomatic recipients received no pre-treatment. Recipient muscles of early symptomatic mice were focally irradiated, and cell transplantation was performed in the presence or absence of cardiotoxin (CTX) injection. **b** Graph depicting the maximum number of dysferlin-expressing, donor-derived fibers per cross-section for each grafted TA muscle. Source data are provided as a Source Data file. **c** Dysferlin immunostaining was performed on transversal cryosections of TA muscles from uninjured, late symptomatic recipients. Dysferlin positive, donor-derived myofibers were found in 1 out of $n = 5$ grafted muscles. Nuclei are stained with Hoechst. Scale bar: 50 μm. **d** Up to 50 dysferlin positive myofibers per section were found in 4 out of $n = 5$ irradiated-only muscles of early symptomatic recipients. Scale bar: 50 μm. **e** Up to 150 dysferlin positive myofibers per section were found in 2 out of $n = 5$ irradiated plus CTX-injured muscles of early symptomatic recipients. Scale bar: 50 μm. **f, g** Pax7-positive cells/nuclei (white arrowheads) were found adjacent to donor-derived, dysferlin expressing myofibers in irradiated-only (**f**) and irradiated + CTX (**g**) recipient muscles. Scale bars: 20 μm. **h** Pax7/laminin immunostaining of the graft in (**e**). The zoomed-in areas contain Pax7-positive cells/nuclei in the MuSC niche (white arrowheads). Nuclei are stained with Hoechst. Scale bars: 50 μm.

Genotyping of *DYSF* exon 44 was performed with primers HE7 + HE8 (Supplementary Table 4).

## Human primary MuSC isolation and culture

Human MuSC isolation was performed as described[55]. Briefly, immediately after the biopsy procedure, the muscle specimen was transferred into Solution A for transport (30 mM HEPES, 130 mM NaCl, 3 mM KCl, 10 mM D-glucose, and 3.2 μM Phenol red, pH 7.6). The fresh muscle specimen was manually dissected into fragments, which were then subjected to hypothermic treatment at 5 °C for 4−7 days prior to downstream processing for MuSC isolation. After hypothermic treatment, fragments were further mechanically dissected and small fragments were cultured in individual vessels in Skeletal Muscle Cell Growth Medium (SMCGM, Provitro) supplemented with 10% FCS in a humidified incubator with 5% $CO_2$, 37 °C, 95% rH to allow outgrowth of oligoclonal MuSC colonies. For passaging, MuSC were washed with Dulbecco's phosphate-buffered saline (DPBS) (Thermo Fisher Scientific) and detached with 0.25% Trypsin-EDTA (Provitro) or TrypLE Express (Gibco) at 37 °C for 5 min. To induce differentiation, MuSC were seeded in IbiTreat-coated 8-well μ-Slides (Ibidi) (5000 to 15,000 cells per well). At 70−80% confluence, the medium was switched to OptiMEM (Thermo Fisher Scientific). After three/four days, cells were processed for laser wounding assay and/or immunostaining.

## mRNA, sgRNA and vectors

SpCas9 and GFP mRNA were purchased from Aldevron. sgRNAs were purchased from Integrated DNA Technologies (IDT). gRNA sequences are listed in Supplementary Table 2. All-in-one plasmids containing a CAG promoter-driven expression cassette for wild-type SpCas9[13] (HE_p4.1) or eSpCas(1.1)[29] (HE_p3.1) followed by T2A-Venus and a human U6 promoter-driven sgRNA expression cassette were previously described[24]. For sgRNA cloning, HE_p4.1 and HE_p3.1 were digested with BpiI, and complementary oligos HE49-HE54 encoding the spacer sequences (Supplementary Table 4) were annealed and ligated. Constructs were verified by Sanger sequencing using primer HE33 (Supplementary Table 4).

## hiPSC generation and characterization

Patient hiPSC were generated and characterized as described[73,74]. All blood-derived and muscle-derived control hiPSC lines used in this study were previously described[24,73,74]. Control fibroblast-derived hiPSC were obtained from The Jain Foundation (https://www.jain-foundation.org/). hiPSC were cultured in mTeSR1 medium (StemCell Technologies) in a humidified incubator with 5% $O_2$, 5% $CO_2$, 37 °C, 95% rH. Cell culture vessels were coated with Matrigel hESC-Qualified Matrix (Corning) following manufacturer's instructions. For maintenance, cells were passaged as clumps using 0.5 mM PBS/EDTA. For transfection or subcloning, cells were detached with Accutase (Thermo Fisher Scientific) to obtain single cell suspensions and medium was supplemented with 10 μM Y-27632 2HCl (Selleckchem,) for 1 day or until colonies consisted of 5−6 cells. Absence of mycoplasma contamination was confirmed using the Venor® GeM qOneStep kit (Minerva Biolabs). Donor identity was confirmed by short tandem repeat (STR) analysis (Supplementary Table 5).

## hiPSC transfection, sorting, and single cell cloning

hiPSC transfection was performed as described[24]. 1.25 μg of HE_p4.1 or HE_p3.1 plasmid DNA were transfected per 300,000 cells using Lipofectamine Stem Transfection Reagent (Thermo Fisher Scientific) according to manufacturer's instructions. For HDR experiments, 3.1−25 pmol of a 120 nucleotide-long single-stranded oligodeoxynucleotide (ssODN) containing the wild-type *DYSF* exon 44 sequence (HE85, Supplementary Table 4) were provided in addition. FACS-sorting to enrich for Venus+ cells was performed 2 days after transfection using a FACSAria Cell Sorter (BD Biosciences) as described[24]. To establish single-cell derived hiPSC colonies, cells were seeded at low density (4000 cells/9.5 $cm^2$) ≥ 7 days after FACS-sorting. Single cell-derived hiPSC colonies were manually picked in aseptic conditions under a laminar flow hood equipped with a microscope and expanded for further analysis.

## Human MuSC nucleofection

Primary human MuSC nucleofection was performed as described[25]. Briefly, cells were harvested using TrypLE Express, centrifuged for 5 min at $200 \times g$, and washed once with DPBS. After a second spin down and supernatant removal, 150,000 cells were resuspended in 20 μl of P5 Primary Cell Nucleofector Solution (Lonza) already containing mRNA and sgRNA when applicable. For 3 μg of SpCas9 mRNA (Aldrevron), 2 μg of 5′/3′-end-modified sgRNA (Integrated DNA Technologies (IDT) or Synthego) (1:0.67 mass ratio) were added per nucleofection reaction. Cells were electroporated with the Amaxa 4D Nucleofector (Lonza) using the X Unit with 16-well nucleofection cuvettes. Next, 80 μl of prewarmed SMCGM were added to each cuvette and cells were transferred to a single well of a 6-well plate containing 2 ml of prewarmed SMCGM. Medium was changed after 24 h.

## Off-target prediction

A total of 81 potential OTS (excluding the wild-type human *DYSF* exon 44) with up to four mismatches to the protospacer were predicted by CRISPOR[28]. We selected the top 10 predicted sites rated by off-target score, regardless of annotation (intergenic, intronic or exonic), plus all predicted exonic OTS for analysis (Supplementary Table 3).

## Genomic DNA (gDNA) extraction and PCR amplification

gDNA was isolated using Agencourt AMPure XP beads (Beckman Coulter). Briefly, each sample was lysed in a heating block with AL-Buffer (Qiagen) containing 0.2 mg/mL Proteinase K (Qiagen) for 10 min at 56 °C. Twice the volume of prewarmed beads was added to each sample and mixed on a rotational wheel. Tubes were placed on a magnetic rack to separate the beads from the supernatant. Beads were washed twice with 80% ethanol, and bound DNA was eluted using FG3 buffer (Qiagen). PCR amplification was performed with Phusion or Q5 High Fidelity DNA Polymerase (New England Biolabs). Primer sequences are provided in Supplementary Table 4.

## On and off-target genome editing analysis

PCR products were purified using the NucleoSpin Gel and PCR Clean-up kit (Machery-Nagel). Sanger sequencing was performed by LGC Genomics and the resulting chromatograms were analyzed with ICE v2.0 (Synthego). Next-generation amplicon sequencing was done at GENEWIZ (Amplicon EZ service) using an Illumina MiSeq platform and 250 bp paired-end reads. Results were analyzed using CRISPResso2[75]. The following parameters were applied: Sequencing design: Paired-end reads; Minimum homology for alignment to an amplicon: 60%; Center of the quantification window (relative to the 3′ end of the provided sgRNA): −3; Quantification window size (bp): 30; Minimum average read quality (phred33 scale): >30; Minimum single bp quality (phred33 scale): No filter; Replace bases with N that have a quality lower than (phred33 scale): No filter; Exclude bp from the left side of the amplicon sequence for the quantification of the mutations: 15 bp; Exclude bp from the right side of the amplicon sequence for the quantification of the mutations: 15 bp.

## Subcloning analysis of PCR amplicons

Subcloning was performed using CloneJET PCR Cloning Kit (Thermo Fisher Scientific) according to the manufacturer's instructions. The ligation mix was transformed into electrocompetent *E. coli* bacteria, which were plated on LB-Ampicillin agar plates and incubated overnight at 37 °C. Single bacteria colonies were picked and analyzed by PCR using the kit's primers and sequenced usingprimers HE7 and HE8 (Supplementary Table 4).

## RNA isolation and quantitative reverse transcription PCR (RT-qPCR)

Total RNA was isolated from cultured cells or tissue cryosections using TRIzol (Thermo Fisher Scientific) following the manufacturer's instructions. cDNA synthesis was performed using the QuantiTect Reverse Transcription kit (Qiagen). RT-qPCR was performed using KAPA SYBR® FAST qPCR Master Mix (2X) Universal (Sigma-Aldrich) following the manufacturer's instructions. Relative mRNA levels were evaluated using CFX Connect Real-Time System (Bio-Rad) with primers SDFp_32 + SDFp_33 (Supplementary Table 4). Ct values were normalized to the housekeeping gene *GAPDH/Gapdh* (DCt values) for each sample. Relative gene expression levels were calculated using the formula $2^{-\Delta\Delta Ct}$ according to the MIQE guidelines.

## GUIDE-seq

GUIDE-Seq experiments were performed as previously described[30] on three control human MuSC populations. In brief, MuSC were nucleofected with SpCas9 mRNA and sgRNA#3 (1:0.67 mass ratio) as well as 100 pmol of a phosphorylated, phosphorothioate-modified dsODN. MuSC nucleofected with SpCas9 mRNA and dsODN, without sgRNA, served as negative control. Cells were harvested 4 days after nucleofection and gDNA was isolated using AMPure beads (Beckmann Coulter). 100 ng of gDNA were tagmented using Tn5 transposase (Illumina), followed by DNA purification using Zymo DNA clean and concentrator kit (Zymo Research). Afterward, GUIDE-Seq amplicons were generated using dsODN tag-specific amplification and Illumina libraries were prepared with specific primer sets (Nextera CD indexes, Supplementary Table 4). GUIDE-Seq libraries were pooled and loaded on Illumina NextSeq500 for NGS.

## Analysis of sequenced GUIDE-seq libraries

GUIDE-seq paired-end reads were aligned to the human genome (hg38) using bwa (Version: 0.7.12-r1039)[76]. Overlapping alignments were clustered per sample and tag-integration sites were identified within the aligned sequences. The clusters of all samples per group (GSP+, GSP+ controls, GSP-, GSP- controls) were then intersected. To exclude false positive clusters, a cut-off of 200 reads per cluster was used. Clusters were then filtered to be positive in at least two GSP+ or

GSP- edited samples and negative in all control samples. The clusters were intersected with the off-target predictions based on CrispRGold v.1.2[31]. The reads of the remaining clusters were then summed per GSP- and GSP+ samples and used for further analysis. As a precaution, a cluster on chromosome 5 (60249111-60249131) was included in the analysis despite being identified in only one sample.

## SDS page and immunoblot

Protein lysates were prepared using RIPA buffer containing protease inhibitors. Protein quantification was performed using the BCA Protein Assay Kit (Pierce) following the manufacturer's protocol. Proteins were separated in denaturing conditions using Invitrogen Novex™ Wedge-Well™ 8–16% gradient Tris–glycine acrylamide gels (Thermo Fisher Scientific) and transferred using a wet electroblotting system (Bio-Rad). Blocking was done with 4% dry milk powder. Primary antibodies (Supplementary Table 6) were applied overnight at 4 °C or for 2 h at room temperature (RT). Incubation with horse radish peroxidase (HRP)-conjugated secondary antibodies (1:5,000) was performed at RT for 1h. For developing, membranes were incubated with Amersham ECL Prime Western Blotting Detection Reagent (GE Healthcare) and imaged with an VWR® CHEMI only system (VWR International). Images were processed using Adobe Photoshop CC 17. Any modifications were applied to the whole image only (containing all the lanes). Western blot densitometry was performed in unprocessed images using ImageJ (NIH).

## Immunostaining of cultured cells

For myogenic marker immunostaining, human and mouse MuSC were plated on IbiTreat-coated 8-well μ-Slides (Ibidi) (8,000–10,000 cells/cm²). After 1 day, they were fixed with a 3.7% formaldehyde solution and permeabilized with 0.2% Triton X-100. For dysferlin immunostaining, cells were allowed to reach density and differentiate, after which they were fixed with ice-cold methanol. Blocking was performed with 1% bovine serum albumin (BSA) in DPBS. Primary antibodies were incubated overnight at 4 °C as indicated in Supplementary Table 6. Alexa Fluor-conjugated secondary antibodies (Invitrogen) were incubated for 1h at RT (1:500 in DPBS). Nuclei were counterstained with Hoechst 33342 (Invitrogen). Images were acquired with a laser scanning confocal microscope LSM 700 (Carl Zeiss Microscopy) and a DMI6000 fluorescence microscope (Leica Microsystems), and processed with ZEN (Carl Zeiss Microscopy), ImageJ and Adobe Illustrator. ≥200 nuclei were counted per sample to calculate percentage values for myogenic and proliferation markers.

## Structural analyses, laser-mediated membrane wounding, and immunostaining

The Ca²⁺-bound C2B domain of rat synaptotagmin-1 (pdb 1TJX)[34] was superimposed with the AlphaFold2-predicted C2F domain of human dysferlin (Expasy accession number: O75923)[32] using *Coot*[77]. Figures were prepared using The PyMOL Molecular Graphics System, Version 2.3.2 Schrödinger, LLC. Human MuSC were plated on IbiTreat-coated 8-well μ-Slides (Ibidi) and induced to differentiate as described above. Before performing the membrane wounding, medium was replaced with Tyrode solution (140 mM NaCl, 5 mM KCl, 2 mM MgCl₂, and 10 mM HEPES, pH 7.2). Myotubes were wounded by irradiation of a 2.5 × 2.5 μm boundary area of the plasma membrane at 100% power (10 mW diode laser, 488, 555, 639 nm) for 76 s using a Zeiss LSM 700 confocal microscope with the 63x objective (LCI Plan-Neofluar 63x/1.3 Imm Korr DIC M27; Zeiss). Confocal images were acquired using the Zeiss LSM ZEN 2.3 software (Carl Zeiss Microscopy) before irradiation and every 20 s after wounding. After laser injury, myotubes were fixed with a 3.7% formaldehyde solution and blocked with 1% BSA/DPBS. Immunostaining with primary antibodies against annexin A1 and dysferlin (Romeo) (Supplementary Table 6) was performed as described above.

## Cloning, expression, and purification of dysferlin's C2F domain

The wild-type C2F domain from human dysferlin isoform 8 (UniProtID: O75923, residues 1575-1792) was amplified from the hDYSF-IRES-GFP plasmid[54] using primers C2F_1575-1792_fwd and C2F_1575-1792_rev (Supplementary Table 4) and cloned into a modified pET21b vector as an N-terminal MBP-tag fusion using Gibson assembly according to the manufacturer's protocol (NEB, # E5510S). The reframed human dysferlin C2F domain was amplified with the same primers as above from cDNA obtained from edited patient hiPSC carrying a homozygous +1A insertion and cloned using the same procedure as described for the wild-type C2F domain. Bacterial cultures were grown in Terrific Broth medium (TB) to an OD600 of 0.6–0.7 at 37 °C followed by a temperature shift to 18 °C. Protein expression was induced at 18 °C by adding 100 μM isopropyl-ß-D-1-thiogalactopyranoside (IPTG) and cultures were grown at 18 °C for another 18 h. Cells were harvested by centrifugation at $5000 \times g$ for 20 min, and stored at −20 °C until use. Upon thawing on ice, cells were resuspended in Buffer A (50 mM HEPES/NaOH pH 7.5, 400 mM NaCl, 5 mM $MgCl_2$, 40 mM Imidazole, supplemented with 2.5 mM β-mercaptoethanol, 6 μg/ml Dnase I, 1 mM PMSF, and Protease Inhibitor Cocktail (Roche, # 11836153001)). Cells were disrupted by passing them through a microfluidizer (Microfluidics). The lysed bacterial suspension was then centrifuged at $46,500 \times g$ for 60 min at 4 °C. The supernatant was collected and immediately applied on a Dextrin Sepharose column equilibrated with buffer B (50 mM HEPES/NaOH pH 7.5, 400 mM NaCl, 40 mM Imidazole, 5 mM $MgCl_2$, 2 mM CaCl2, 2.5 mM β-mercaptoethanol). This was followed by extensive washing using buffer B. Protein was eluted with buffer B, containing 20 mM maltose. To remove remaining contaminations, protein was loaded on Superdex 200 gel-filtration column equilibrated with buffer C (20 mM HEPES/NaOH pH 7.5, 300 mM NaCl, 2.5 mM $MgCl_2$, 2.5 mM $CaCl_2$, and 2.5 mM Dithiothreitol (DTT)). Fractions containing pure protein were pooled and used directly for the assay. For the ITC experiment, samples were incubated with 5 mM EDTA to remove metal ions, and subsequently, EDTA was removed using a buffer exchange HiTrap column equilibrated with buffer D (20 mM HEPES/NaOH pH 7.5, 150 mM NaCl, and 2.5 mM DTT).

## Thermal shift assay (TSA)

TSA was conducted similarly to Mohd et al.[78]. Briefly, proteins (~10 μM) were mixed with a fluorescent dye (ThermoFisher Scientific, # 4461146) in a final volume of 20 μl, transferred to a 96-well PCR plate and then sealed with a plastic film. The reactions were kept on ice. The fluorescence was measured from 20 °C to 100 °C with 1 °C temperature steps using a CFX96 touch-real time PCR (Bio-Rad).

## Isothermal titration calorimetry (ITC)

ITC experiments were carried out as described[79]. Briefly, titration was performed using an iTC200 Isothermal Titration Calorimeter (iTC200 system, MicroCal™, GE Healthcare, Freiburg). A 150 mM $CaCl_2$ solution in ITC Buffer (50 mM HEPES/NaOH, pH 7.5, 150 mM NaCl, 2.5 mM DTT) was titrated in 2 μL steps into a reaction chamber containing 10 or 14 μM MBP-C2F in the same buffer at 18 °C. The resulting heat change upon injection was integrated over a time range of 45 min, and the obtained values were fitted to a standard single-site binding model using Origin®.

## Animal experiments

Mice were handled according to institutional guidelines under experimentation licenses approved by the LaGeSo (Berlin, Germany) and housed in individually ventilated cages in a specific pathogen-free facility with a 12 h light/dark cycle, $22 \pm 2$ °C ambient temperature, $55 \pm 10\%$ ambient humidity, and with ad libitum access to food and water. Cages were provided with nest-building material and shelter. Animal experiments complied with local animal welfare regulations and reporting abides by the ARRIVE guidelines.

## Generation of hEx44mut and hEx44wt mice

Mutant mice were produced by microinjection of C57BL/6N zygotes as described[80] using Cas9 protein (IDT), synthetic guide RNA (IDT) targeting *Dysf* exon 44 (ACTTTCCGAATACAGGCTCC) and a targeting vector as recombination template. Targeting vectors containing either the wild-type or mutant human *DYSF* exon 44 sequence, flanked by 1,400 bp-long sequences homologous to the mouse *Dysf* locus flanking the murine *Dysf* exon 44 (5′ and 3′) were purchased as GeneArt synthetic fragments readily inserted into the pMX vector backbone (Thermo Fisher Scientific) (Supplementary Fig. 8c, d). The complete inserts were verified by Sanger sequencing using primers HE14-HE17 (Supplementary Table 4). For microinjections, zygotes were obtained by mating C57BL/6N males with superovulated C57BL/6N females (Charles River). Zygotes were injected into one pronucleus and transferred into pseudo-pregnant NMRI female mice to obtain live pups. All mice showed normal development and appeared healthy. F0 animals were genotyped by PCR-RFLP using primers HE12 + HE13 (Supplementary Table 4), followed by enzymatic digestion with BanII (hEx44wt allele) or AvaII (hEx44mut allele) (Supplementary Fig. 8e, f). All RFLP-positive founders were analyzed by Sanger sequencing (Supplementary Fig. 8g, h). For *hEx44wt*, 2/15 F0 animals harbored a heterozygous knock-in, although in both cases not all the mismatched positions between the mouse and human wild-type exon 44 were exchanged and one founder with all but one exchanged positions was selected (Supplementary Fig. 8i). For *hEx44mut*, 10/31 F0 animals harbored a heterozygous knock-in or were mosaic and heterozygous animals harboring the correct mutant *DYSF* exon 44 knock-in sequence (Supplementary Fig. 8j) were selected. To establish homozygous colonies, heterozygous F0 were crossed to wild-type C57BL/6N mice and heterozygous F1 progeny was inbred to obtain homozygous F2 mice. Homozygous F2 progeny deriving from each F0 mouse was further analyzed by Sanger sequencing of the *Dysf* locus flanking exon 44, from outside the homology region of the targeting vectors using primers HE223-HE226 (Supplementary Table 4). The progeny from one founder harboring an intact *hEx44wt* or *hEx44mut* knock-in allele was selected to establish homozygous colonies for each transgenic line.

## Mouse primary MuSC isolation, culture and nucleofection

MuSC were isolated from 4- to 12-week-old homozygous $Dysf^{hEx44wt/hEx44wt}$ and $Dysf^{hEx44mut/hEx44mut}$ mice. MuSC from homozygous hEx44wt mice were utilized as positive control for dysferlin expression analysis. For transplantation into $Dysf^{hEx44mut/hEx44mut}$ recipients, siblings were used as MuSC donors whenever possible. MuSC were isolated from hind limbs as described[81]. Briefly, mice were euthanized by cervical dislocation and muscle tissue was dissected, mechanically minced, and digested with NB4 collagenase (12 mg/ml) (SERVA) and Dispase II (100 U/ml) (Roche) for 1 h, and TrypLE Express (Gibco) for 5 min. The resulting cell suspension was stained with PE-conjugated anti-Sca1, -CD31, and -CD45 antibodies and an anti-VCAM1 primary antibody (Supplementary Table 6) plus a secondary Alexa Fluor 488-conjugated secondary antibody. Sca1(neg), CD31(neg), CD45(neg), VCAM1(pos) cells were selected with a FACSAria Cell Sorter (BD Biosciences) (Supplementary Fig. 14a). Pax7/Desmin immunostaining was performed as above to determine purity (Supplementary Fig. 14b, c). Isolated cells were plated on Laminin- (3 μg/cm²) (Millipore) coated dishes. Cells were cultured in DMEM/F12 enriched with 15% FCS, bFGF (2.5 μg/ml) (Sigma Aldrich) and B27 supplement without vitamin A (Gibco) (mouse MuSC medium) at 37 °C in hypoxia in a humidified incubator with 5% $CO_2$, 5% $O_2$, 37 °C, 95% rH. For passaging, cells were washed with DPBS and detached with TrypLE Express (Gibco) at 37 °C for 5 min. Mouse primary MuSC nucleofection was performed as described above. 2.5 μg of SpCas9 mRNA (Aldevron) plus 1.7 μg of 5′/3′-end-modified sgRNA (IDT or Synthego) (1:0.67 mass ratio) were added to 150,000 cells per nucleofection reaction.

## Mouse MuSC transplantation

The term autologous is used when MuSC, isolated from the inbred mouse strain (*Dysf* $^{hEx44mut/hEx44mut}$) are transplanted into the muscle of the same mouse strain. Early symptomatic (11- to 14-week-old, $n = 5$ males) or late symptomatic (75-week-old, $n = 5$ males/females) homozygous hEx44mut (*Dysf* $^{hEx44mut/hEx44mut}$) mice were used as recipients. MuSC isolated from homozygous hEx44mut mice were nucleofected as described above 2–4 days prior to transplantation and cultivated in mouse MuSC medium enriched with 15% mouse serum (instead of FCS) for at least 2 days before transplantation. Early symptomatic mice received a cell transplantation in both TA muscles. As pre-treatment, one muscle was irradiated only whilst the contralateral muscle was irradiated and injected with CTX. In late symptomatic mice, cell transplantation was applied to one TA muscle and the contralateral muscle was used as control. In early symptomatic mice, 16-Gy focal irradiation of recipient hind limbs was performed 1 day prior to cell transplantation as described[82]. 15 μL containing 50,000-100,000 cells in a sterile PBS + 5% mouse serum solution were injected using a 25 μL, model 702 RN SYR Hamilton Syringe coupled to a custom-made 20-mm long 26 g small hub removable needle into the medial portion of the TA muscle following a longitudinal direction as described. When CTX was applied, 40 μl of a 10 μM CTX/PBS solution (Latoxan) were injected with a 26-gauge hypodermic needle in the median portion of the TA muscle 5 min before the cell injection. Mice were euthanized by cervical dislocation 3 weeks after transplantation and grafted muscles were cryopreserved for further analysis.

## Histological analysis of mouse muscles

Mice were euthanized by cervical dislocation at the indicated time points. Mouse muscles were embedded in gum tragacanth, cryopreserved in liquid nitrogen-chilled isopentane, frozen in liquid nitrogen, and stored at −80 °C. 6-μm-thick transversal cryosections were cut with a CM3050 S cryostat (Leica). For dysferlin immunostaining, sections were fixed for 5 min in acetone at −20 °C, blocked with 1% BSA/PBS, and incubated overnight at 4 °C with an anti-dysferlin (Romeo) antibody (Supplementary Table 6). For Pax7/laminin and Pax7/dysferlin immunostaining, 6-μm cryosections were fixed in 3.7% formaldehyde, permeabilized with 0.2% Triton-X, blocked with 5% BSA plus 3% goat serum in PBS followed by a blocking with M.O.M.® (Mouse on Mouse) Blocking Reagent (Vector Laboratories), and incubated overnight at 4 °C with anti-Pax7 and anti-laminin antibodies (Supplementary Table 6). For eMyHC, F4/80, CD3, CD8α, and CD20 immunostainings, 6-μm cryosections were fixed in 3.7% formaldehyde, permeabilized with 0.2% Triton-X, blocked with 1% BSA in PBS and incubated overnight at 4 °C with the corresponding antibodies (Supplementary Table 6). Alexa Fluor-conjugated secondary antibodies were applied for 2 h at RT in DPBS (1:500). Gomori's one-step trichrome and Picrosirius red stains were performed following standard protocols. Confocal images were acquired using Zeiss LSM 700 or LSM 900 microscopes (Carl Zeiss Microscopy). Bright field images of histological stains were acquired using a Leica DM LB2 microscope (Leica Microsystems). Images were processed using ZEN (Carl Zeiss Microscopy), ImageJ (NIH) and Adobe Illustrator. For the quantitative phenotypic analysis by Gomori's trichrome staining, three 20x images of randomly selected areas were acquired per muscle with a Leica DM LB2 microscope and manually analyzed using the ImageJ cell counter plugin. For the calculation of minimum Feret diameters, low magnification images of laminin immunostaining were acquired with an LSM 900 microscope and analyzed using Myosoft[83].

## Statistics

All experiments were performed in at least three biologically different replicates unless otherwise indicated in the figure legend. Details about the group size and statistical tests are described in the corresponding figure legend. All statistical analysis and graphs were performed using GraphPad Prism Software (version 8.0). Graphs show the mean ± SD where applicable.

## Reporting summary

Further information on research design is available in the Nature Portfolio Reporting Summary linked to this article.

## Data availability

All raw NGS data generated in this study plus corresponding metadata are available at the NCBI Sequence Read Archive (SRA) under the Bioproject accession number PRJNA1175452. Source data are provided with this paper.

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

## Acknowledgements

The study was funded by the Stiftung Gisela Krebs through a grant to H.E. J.S. and O.D. were funded by the Deutsche Forschungsgemeinschaft (grant no. TRR186/A23 to O.D.). We thank the patients and all muscle biopsy donors who make our work possible. We thank Andreas Spuler for providing biological specimens. We thank Stefanie Haafke, Stephanie Meyer-Liesener, Anne Krause, Adrienne Rothe and Denise Rossa for excellent technical assistance. We thank Busem Ignak for helping with transplantation experiments, and Leon Kersting and Gracia Peralta for helping with Myosoft analysis. We thank Sebastian Diecke and the MDC iPSC core facility for helping with iPSC generation and characterization. We thank Hans Peter Rahn for technical assistance with FACS-sorting.

## Author contributions

H.E. and S.dF. designed, conducted, and analyzed experiments. J.S. designed, conducted and analyzed biophysical experiments. R.G. designed and analyzed GUIDE-seq. S.M. performed GUIDE-seq and helped with off-target validation by NGS. A.M. isolated human MuSC. S.K., A.M., and M.P. contributed to the analysis of the transplantation experiments. A.Z. performed mouse phenotyping. E.M. provided control hiPSC and conducted experiments. O.D. performed structural analysis. R.K. generated transgenic mice. H.E., O.D., and Si.S. discussed the results. H.E., S.dF., and J.S. prepared figures. H.E. and Si.S. designed the study and coordinated the project. H.E. and Si.S. wrote the manuscript with contributions from S.dF. J.S. and O.D.

## Funding

## Competing interests

Si.S. is an inventor on a technology for primary human muscle stem cell isolation and manufacturing (IP: (DE10 2014 216872), 2015 PCT (WO 2016/030371), granted in EU and US). Si.S. and H.E. are co-inventors on a pending patent application on gene editing of human muscle stem cells (European Patent Office 21 160 696.7). Si.S. is co-founder of MyoPax GmbH and MyoPax Denmark ApS. The remaining authors declare no competing interests.
