## [Transparent Peer Review file · Nature Communications]

Gene-edited primary muscle stem cells rescue dysferlin expression in muscular dystrophy across species boundaries

Corresponding Author: Dr Helena Escobar

Version 1:

Reviewer comments:

Reviewer #1

(Remarks to the Author)

Escobar et al demonstrate that gene editing of muscle stem cells followed by muscle stem cell transplantation can result in restored dysferlin expression to myofibers. The manuscript combines two lines of inquiry. First, they demonstrate expression of dysferlin in iPSCs and show that frame-shift editing of patient-derived iPSCs can rescue full-length dysferlin protein expression. In a second line of inquiry, the authors describe a *tour-de-force* experiment, in which they generate a humanized mouse that harbors the patient mutation and has a loss of dysferlin expression. They then go on to show that frame-shift editing of muscle stem cells can result in restored dysferlin expression in stem cell derived myofibers *in vivo*. The results are important to the field and of general interest given its approach of combining patient-derived stem cells and a humanized disease model. Moreover, the demonstration that changing the reading frame through gene editing can be of translational value is of wide interest. Nevertheless, there are a number of points where the manuscript needs to be strengthened before publication.

-The story deserves a better characterization of the novel mouse model. The authors describe using early or late symptomatic mice but fail to define the symptoms used to define these categories. Do the mice show a force deficit? Do the mice recapitulate stronger phenotypes in proximal hind limb muscles (ie quad), as in *sjl* mice?

- Central-nucleated fibers are a hallmark of regeneration, but insufficient to demonstrate dystrophy. Please include quantification of specific regeneration markers (ie embryonic MHC) and immune infiltration markers (ie F4/80).

-Figure 8: While the images are compelling, not all transplants resulted in a detectable graft (nicely explained in legend). Why was the transplantation more effective in the uninjured condition (in terms of the number of positive grafts) compared to the injured condition? The number of grafted myofibers also seems low given the large quantity of cells transplanted (100,000). This suggests a variability in the transplantation assay, which complicates the interpretation and should be addressed.

-Has the dystrophy been rescued, as the title promises? The authors do an excellent job of showing restoration of Dysferlin expression on myofibers. However, whether this is sufficient to rescue disease phenotypes remains unclear. The authors should provide better characterization of the mice following transplantation and show that the muscle is indeed better.

Minor comments:

- A purity of 50% Pax7+ MuSCs could indicate suboptimal stem cell isolation, potentially contributing to the variable engraftment.

-Figure 5c. It appears that the dysferlin null cells can repair the laser-inflicted membrane damage. Is that correct? If so, is this assay a good readout for Dysferlin function (as it shows trafficking only)?

-Page 4, line 87 and discussion. Can it be called "autologous" when the donor and recipient are not the same individual?

This wording needs to be addressed.

- Supp fig 12: please also show lower magnification images to illustrate the effects of the corrected MuSCs by revealing mutant fibers without dysferlin staining.

Reviewer #2

(Remarks to the Author)

This paper explores the potential of CRISPR-mediated gene editing to address muscular dystrophy caused by dysferlin deficiency. While it offers valuable insights, it faces several critical points that need addressing.

Firstly, the study's duration for assessing the effects of gene-edited cells post-transplantation is notably limited, only evaluating muscle outcomes at three weeks post-engraftment. This short-term evaluation raises concerns about the longevity and stability of the corrected phenotype. Long-term observations are crucial to understanding the sustained efficacy and potential risks associated with these edited cells.

Additionally, the paper lacks detailed functional assays to comprehensively evaluate the restoration of muscle function and strength in the treated mice. Without these assessments, the true therapeutic potential of the gene-edited cells remains ambiguous.

Moreover, the paper's evidence regarding the ability of the rescued dysferlin to repair membranes lacks robustness. The presented data show faint signals at the injury site, casting doubts on the efficacy of the membrane repair process. Incorporating staining techniques like Evans Blue could provide a clearer *in vivo* understanding of the efficacy of membrane repair after muscle injury.

The study fails to thoroughly investigate the immune response to the edited cells or the potential immunogenicity of the gene-editing tools. This oversight is crucial, especially considering the prospects of clinical translation. Understanding immune reactions and potential risks of rejection or adverse reactions is imperative for safe and effective therapeutic applications. Expanding the assessment duration would significantly enhance the depth and reliability of our analysis. Furthermore, while the paper acknowledges the assessment of potential off-target sites, it lacks extensive analysis or validation of these sites. A more comprehensive analysis would bolster the study's conclusions regarding the specificity and safety of the gene-editing approach, ensuring minimal off-target effects. The authors suggest that a pipeline should be used, but how will this tool be able to analyze the off-target effects that will certainly be present after the modifications and ensure that only the intended modification is present? A low frequency is not a null frequency, so is the risk of these modifications spreading during divisions not null as well.

Finally, the creation of four missense mutations, particularly the modification at the 1624 position, raises concerns highlighted by predictive tools. The predictive pathogenicity of the p.Glu1624Gly mutation suggests potential deleterious effects, indicating a risk that may compromise the intended therapeutic outcomes - see UMD predictor in UMD dysf database for example (76/100 pathogenic score) - . Relying solely on the observation of the predicted structure isn't sufficient to ensure the absence of deleterious effects, much like the absence of mutations observed in patients.

In summary, while the paper demonstrates promise in addressing dysferlin deficiency using CRISPR-mediated gene editing, it falls short in various critical aspects. Addressing these limitations is vital to ensure the safety, efficacy, and long-term viability of this therapeutic approach before clinical translation.

Minor points:

Line 38 and others : LGMD2B is now known as LGMDR2 please consider this change.

Line 42 : tandem C2: I can't find a reference to explain why this C2 domain are called tandem ? do you have one pertinent reference for this?

On line 61, even though dysferlin may not inherently possess redundant domains, existing references indicate that certain domains of dysferlin are deemed less critical.

Refs : doi:10.1016/j.ymthe.2017.05.013 ; doi: 10.1126/scitranslmed.3000951 for example

Line 163 : two spaces between We generated.

Line 189 : the reticular pattern is not clearly seen in the image provided.

Line 217 : It seems somewhat strange that these authors do not acknowledge the potential for exon skipping in dysferlin, especially considering their publication using this technique and the number of articles demonstrating its efficacy.

Refs : doi: 10.1007/978-1-0716-2772-3_11; doi: 10.1089/nat.2019.0788 ; doi: 10.1016/j.omtn.2018.08.013 ; doi: 10.3233/JND-150109 ; 10.1093/hmg/ddv141.

Line 222 : The database suggests that only 38 patients might benefit from this approach, indicating a limited number of potential beneficiaries.

Line 230 : the mutation V67D in the C2A is also quite well explored on its effect on protein structure and ions binding.

Line 232 : The systematic assessment of splicing defects caused by missense mutations has been thoroughly investigated in the past see: doi: 10.1002/humu.22710.

Line 313: I was unable to locate the specific experiment addressing clonal drift.

Line 456 : There is no experiment provided that demonstrates time-dependent membrane repair.

Line 493: it is not clear if the MuSC are obtained from 4- to 12-week -old homozygous Dysf hEx44wt/hEx44wt or from sibling line 513?

Line 544 : The statistical methods employed in this study may not be entirely suitable due to the notably small group sizes. For instance, using mean and standard deviation might not adequately represent the data due to its limited sample size.

Reviewer #3

(Remarks to the Author)

In the paper titled "Gene-edited primary muscle stem cells rescue dysferlin-deficient muscular dystrophy" by Escobar et al., the authors successfully showcase the restoration of dysferlin function in cells afflicted with a prevalent mutation in the DYSF gene through gene editing. Despite the mutation-specific nature of the strategy, the study affirms its efficacy in both in vitro and in vivo settings, particularly in a humanized model of the disease. The paper is commendably written, with clear presentation of data. My primary suggestion pertains to a substantial comment regarding the in vivo quantification of the recovery, along with a few minor suggestions.

Major

In the section titled "Exon 44 re-framed murine MuSC regenerate muscle and rescue dysferlin in hEx44mut mice," I recommend the inclusion of a graph depicting the count of positive myofibers alongside the quantification of restoration efficiency through western blot analysis. This addition would significantly enhance the clarity of the differences observed among the three pretreatment regimens and provide a more comprehensive understanding of the overall in vivo efficacy of the proposed strategy.

Minor

Line 111: reference for Fig 2b is missing in the main text.

Line 114: define DSB.

SpCas9 is sometimes in italics and sometimes not: please be consistent.

Fig 2g: quantify the dysferlin recovery in the 2 clones compared with the hPSCs controls.

Line 164: reference for Fig 6a is missing in the main text.

Discussion: the long-term in vivo effect of the proposed treatment is not discussed. Please provide the authors' view about this topic.

Version 2:

Reviewer comments:

Reviewer #1

(Remarks to the Author)

Escobar et al. have submitted a revised version of their manuscript. In it, they demonstrate restoration of dysferlin expression following transplantation of gene-corrected muscle stem cells. They have now included a more detailed analysis of the muscle phenotype post-treatment and better assessment of off-target effects of their approach. Moreover, they include biophysical analysis of the protein domain encoded by the reframed exon. These new data significantly strengthen the manuscript and address all of my previous comments.

Reviewer #2

(Remarks to the Author)

Dear authors,

Thank you for your effort to answer all questions and concerns raised during the first round of revision. I am now satisfied with your reply and I think the manuscript is well-improved and is now suitable for publication.

Reviewer #3

(Remarks to the Author)

The authors have addressed all the points I raised in the revision. I agree with the publication of the paper.

REVIEWER COMMENTS

Reviewer #1 (Remarks to the Author):

Escobar et al. demonstrate that gene editing of muscle stem cells followed by muscle stem cell transplantation can result in restored dysferlin expression to myofibers. The manuscript combines two lines of inquiry. First, they demonstrate expression of dysferlin in iPSCs and show that frameshift editing of patient-derived iPSCs can rescue full-length dysferlin protein expression. In a second line of inquiry, the authors describe a tour-de-force experiment, in which they generate a humanized mouse that harbors the patient mutation and has a loss of dysferlin expression. They then go on to show that frameshift editing of muscle stem cells can result in restored dysferlin expression in stem cell derived myofibers *in vivo*. The results are important to the field and of general interest given its approach of combining patient-derived stem cells and a humanized disease model. Moreover, the demonstration that changing the reading frame through gene editing can be of translational value is of wide interest. Nevertheless, there are a number of points where the manuscript needs to be strengthened before publication.

1. The story deserves a better characterization of the novel mouse model. The authors describe using early or late symptomatic mice but fail to define the symptoms used to define these categories. Do the mice show a force deficit? Do the mice recapitulate stronger phenotypes in proximal hind limb muscles (ie quad), as in *sjl* mice?

We now provide a thorough characterization of the histological alterations, including older mice and proximal muscles (Supplementary Fig. 9 and 10). The correlation of histological and functional decline is well described for muscular dystrophy patients. In mouse models, the correlation varies (doi: 10.1186/s40478-022-01354-3; doi: 10.1016/S0960-8966(98)00114-X; doi: 10.1038/s41598-019-50550-0; doi: 10.3892/etm.2021.10042). However, a slowly progressing histopathological and clinical phenotype is common to dysferlin-deficient mouse models (doi: 10.1016/j.nmd.2013.02.004; doi: 10.1016/j.omtn.2018.08.013; doi: 10.3390/biomedicines11051438). In future studies of systemic *in vivo* gene editing in hEx44mut mice, clinical assessment of treated and untreated mice will be performed.

2. Central-nucleated fibers are a hallmark of regeneration, but insufficient to demonstrate dystrophy. Please include quantification of specific regeneration markers (ie embryonic MyHC) and immune infiltration markers (ie F4/80).

We now provide stainings of eMyHC and immune cells in 40-week-old hEx44 quadriceps (Supplementary Fig. 12 and 13). Few clusters of eMyHC-positive fibers are found in hEx44mut mice at 40 weeks. Macrophage infiltration is very prominent in hEx44mut muscles (Supplementary Fig. 12). We found a few clusters of CD3+ T-lymphocytes. CD8- and CD20-positive cells were scarce and occurred isolated (Supplementary Fig. 13). No lymphocytes were identified in age- and gender-matched control hEx44wt muscles.

3. Figure 8: While the images are compelling, not all transplants resulted in a detectable graft (nicely explained in legend). Why was the transplantation more effective in the uninjured condition (in terms of the number of positive grafts) compared to the injured condition? The number of grafted myofibers also seems low given the large quantity of cells transplanted (100,000). This suggests a variability in the transplantation assay, which complicates the interpretation and should be addressed.

We now address this in the discussion (*lines 313-316 / 323 of the revised manuscript with marked changes*). We believe that the variability in the transplantation outcomes in CTX-treated muscle is due to technical limitations of a single cell injection and the fact that the effect of the CTX, although widespread, did often not reach the entirety of the muscle's CSA. Hence, the host environment where the donor cells are transplanted may vary. In addition, to perform all injections in a single surgery session and hence minimize the stress for the animals, a relatively large volume of the CTX solution (40 μ L) was injected shortly before the cell suspension. This produces swelling of the small TA muscle

and may influence the distribution of the cells injected later into the tissue in a relatively small suspension volume. In irradiated-only muscles, donor fibers were consistently found along the injection trajectory, coinciding with previous results (Escobar et al., 2016 *Mol Ther – Nucleic Acids*; Marg et al., 2019 *Nat Commun*; Escobar et al., 2021 *JCI Insight*). Also, according to our previous data, donor myofibers were found only in areas of the host muscle that are actively undergoing regeneration (e.g. the needle trajectory or the CTX-damaged area), which limits the extent of engraftment even if a large number of MuSC is injected. We believe that those issues could be addressed clinically by applying multiple closely spaced small-volume injections with high-precision surgical equipment (so-called high-density technique), as performed in allogeneic transplantation trials for DMD (Skuk & Tremblay, 2014, *Mol Ther*, DOI: 10.1038/mt.2014.57).

4. Has the dystrophy been rescued, as the title promises? The authors do an excellent job of showing restoration of Dysferlin expression on myofibers. However, whether this is sufficient to rescue disease phenotypes remains unclear. The authors should provide better characterization of the mice following transplantation and show that the muscle is indeed better.

Please see our answer to question 1. After careful consideration, we decided that performing isolated muscle force measurements is likely not going to be informative: a) because the single muscle force measurements have multiple technical hurdles that interfere with the interpretation of the result; b) because several hundred newly built muscle fibers may not become dominant in the performance of the entire muscle; and c) because in muscles that have been irradiated and injured with CTX, force measurements do not reflect the physiological potential in future patients. We show for the first time rescue of dysferlin expression *in vivo* using a CRISPR-based approach, and we, for the first time, show that autologous gene repaired muscle stem cells engraft, regenerate muscle, repopulate the stem cell niche and rescue the expression of a muscular dystrophy-related protein.

Minor comments:

- A purity of 50% Pax7+ MuSCs could indicate suboptimal stem cell isolation, potentially contributing to the variable engraftment. Our isolation and gating strategy yielded 100% Desmin+ myogenic cell populations as previously described (Bröhl et al., 2012, doi: 10.1016/j.devcel.2012.07.014). Pax7/Desmin immunostaining was performed one day after, and Pax7 expression is known to be quickly downregulated in mouse MuSC after isolation. We have previously shown that Pax7 expression in *ex vivo* cultured human MuSC is not a determinant of their *in vivo* engraftment potential (Marg et al., 2019, doi: 10.1038/s41467-019-13650-z). Whether the same is true for mouse MuSC remains to be seen.
- Figure 5c. It appears that the dysferlin null cells can repair the laser-inflicted membrane damage. Is that correct? If so, is this assay a good readout for Dysferlin function (as it shows trafficking only)? We are not aware of any previous study showing that Annexin A1 does not localize at the injury site in the absence of dysferlin. Indeed, the fact that reframed dysferlin is enriched at the injury site indicates proper trafficking, but we agree that the membrane wounding assay is a suboptimal assay to evaluate functionality, as also agreed by the dysferlin community. However, a better assay has not been developed. We have now thus focused on investigating the influence of the four amino acid exchanges on the stability and function of the C2F domain and we provide new biophysical data showing that reframing in the context of the isolated C2F domain does not majorly affect its thermal stability and calcium binding affinity (Supplementary Fig. 7). Thus, at least on the level of the C2F domain, where the mutations are localized, reframed dysferlin appears functional.
- Page 4, line 87 and discussion. Can it be called “autologous” when the donor and recipient are not the same individual? This wording needs to be addressed. We added the following sentence to Material and Methods (*lines 630-631 of the revised manuscript with marked changes*): “The term autologous is used when MuSCs, isolated from the inbred mouse strains, are transplanted into muscle of the same mouse strain.”

- Supp fig 12: please also show lower magnification images to illustrate the effects of the corrected MuSCs by revealing mutant fibers without dysferlin staining. Please see the new data in Supplementary Fig. 16c.

Reviewer #2 (Remarks to the Author):

This paper explores the potential of CRISPR-mediated gene editing to address muscular dystrophy caused by dysferlin deficiency. While it offers valuable insights, it faces several critical points that need addressing.

1. Firstly, the study's duration for assessing the effects of gene-edited cells post-transplantation is notably limited, only evaluating muscle outcomes at three weeks post-engraftment. This short-term evaluation raises concerns about the longevity and stability of the corrected phenotype. Long-term observations are crucial to understanding the sustained efficacy and potential risks associated with these edited cells. Additionally, the paper lacks detailed functional assays to comprehensively evaluate the restoration of muscle function and strength in the treated mice. Without these assessments, the true therapeutic potential of the gene-edited cells remains ambiguous.

In previous studies (Escobar et al., 2016 *Mol Ther – Nucleic Acids*) we did not observe differences in short- vs long-term engraftment. After careful consideration, we decided that performing isolated muscle force measurements is likely not going to be informative: a) because the single muscle force measurements have multiple technical hurdles that interfere with the interpretation of the result; b) because several hundred newly built muscle fibers may not become dominant in the performance of the entire muscle; and c) because in muscles that have been irradiated and injured with CTX, force measurements do not reflect the physiological potential in future patients. We show for the first time that CRISPR-edited MuSC engraft, regenerate muscle, repopulate the stem cell niche and rescue dysferlin expression in an autologous transplantation context. Further refinement of the transplantation technique will be required to evaluate efficacy in a preclinical and clinical context in a meaningful manner. We have now further elaborated on safety, efficacy and immunological aspects in the discussion (*lines 323-333/374 of the revised manuscript with marked changes*).

2. Moreover, the paper's evidence regarding the ability of the rescued dysferlin to repair membranes lacks robustness. The presented data show faint signals at the injury site, casting doubts on the efficacy of the membrane repair process. Incorporating staining techniques like Evans Blue could provide a clearer in vivo understanding of the efficacy of membrane repair after muscle injury.

Performing laser injury experiments on primary patient myotubes after genetic manipulation *ex vivo* is technically challenging. However, we could show that reframed dysferlin is enriched at the injury site, indicating proper trafficking. Truncated dysferlin variants lacking entire protein domains have been shown to be active in membrane repair assays (e.g. Krahn et al., 2010, doi: 10.1126/scitranslmed.3000951; Malcher et al., 2018, doi: 10.1016/j.omtn.2018.08.013; Lee et al., 2018, doi: 10.1016/j.omtn.2018.10.004; Muriel et al., 2024, doi: 10.1016/j.omtm.2024.101257). However, several of these variants do not fulfill other functions of the full-length protein or are able to rescue the muscle pathology (Lostal et al., 2012, doi: 10.1371/journal.pone.0038036; Muriel et al., 2024, doi: 10.1016/j.omtm.2024.101257). Therefore, it is doubtful whether membrane repair outcomes are a good predictor of the therapeutic potential of a dysferlin variant. Hence, we have focused on investigating the influence of the four amino acid exchanges on the stability and function of the C2F domain. We now provide new biophysical data showing that reframing in the context of the isolated C2F domain does not majorly affect its thermal stability and calcium binding affinity (Supplementary Fig. 7). Thus, at least on the level of the C2F domain, where the mutations are localized, reframed dysferlin appears functional.

3. The study fails to thoroughly investigate the immune response to the edited cells or the potential immunogenicity of the gene-editing tools. This oversight is crucial, especially considering the prospects

of clinical translation. Understanding immune reactions and potential risks of rejection or adverse reactions is imperative for safe and effective therapeutic applications. Expanding the assessment duration would significantly enhance the depth and reliability of our analysis.

By the time of transplantation, the gene editing tools are degraded. A time course of the presence of gene editing tools in MuSC edited using mRNA-mediated delivery is provided in Müthel et al., 2023 (doi: 10.1016/j.omtn.2023.02.005). Immunogenicity of the gene editing tools is therefore not an issue in our approach. We now provide immunostains of immune markers on the grafted tissue (Supplementary Fig. 17). An in-depth analysis of the crosstalk between the donor cells and the host muscle, e.g. by single cell or spatial transcriptomics, will be more informative and is an interesting future prospect. We have now further elaborated on immunological aspects in the discussion (*lines 325-333 of the revised manuscript with marked changes*).

4. Furthermore, while the paper acknowledges the assessment of potential off-target sites, it lacks extensive analysis or validation of these sites. A more comprehensive analysis would bolster the study's conclusions regarding the specificity and safety of the gene-editing approach, ensuring minimal off-target effects. The authors suggest that a pipeline should be used, but how will this tool be able to analyze the off-target effects that will certainly be present after the modifications and ensure that only the intended modification is present? A low frequency is not a null frequency, so is the risk of these modifications spreading during divisions not null as well.

We have now performed GUIDE-seq analysis for unbiased off-target nomination and validated the newly identified off-target sites by next-generation amplicon sequencing (Fig. 3e, f). Overall, the off-target profile of this gRNA is very benign according to both *in silico* predictions and empirical data.

5. Finally, the creation of four missense mutations, particularly the modification at the 1624 position, raises concerns highlighted by predictive tools. The predictive pathogenicity of the p.Glu1624Gly mutation suggests potential deleterious effects, indicating a risk that may compromise the intended therapeutic outcomes - see UMD predictor in UMD dysf database for example (76/100 pathogenic score). Relying solely on the observation of the predicted structure isn't sufficient to ensure the absence of deleterious effects, much like the absence of mutations observed in patients.

The human splicing finder (HSF), which is integrated into the UMD predictor tool, predicts consequences of mutations affecting existing splice signals (donor and acceptor sites, branchpoints and cis-acting elements such as exonic splicing enhancers and silencers) or creating novel ectopic splicing sequences, therefore affecting the function of dysferlin. However, we have convincingly shown that the introduced sequence change does not lead to alterations of splicing, e.g. the protein is abundant and produced as full-length protein. On the other hand, 'UMD-Predictor combines data such as localization within the protein, conservation and biochemical properties of the mutant and wild-type residues, as well as results from HSF analysis to calculate a pathogenicity score ranging from 0 to 100 for each missense variant (score >65 indicates a probable or highly likely pathogenicity).' In fact, also AlphaFold2 employs extensive sequence conservation analyses together with a huge database of existing structures and a highly sophisticated AI-based algorithm to predict, not only the position/biochemical property of an amino acid exchange, but the entire 3D structure of the protein. To infer consequences of mutations based on an AlphaFold2-structural prediction appears therefore more direct and accurate than using the less well-defined pathogenicity score of UMD-predictor. To address this issue also experimentally, we recombinantly expressed and purified the wild-type and reframed C2F domain as maltose-binding protein (MBP-) fusions (new Supplementary Fig. 7a). Thermal shift experiments indicated that the mutations in the C2F domain do not affect the overall stability of the domain (new Supplementary Fig. 7b, c). Furthermore, isothermal titration calorimetry experiments point to similar Ca²⁺-binding affinities of the wild-type and reframed C2F domains (new Supplementary Fig. 7d). These data argue against a major effect of the mutations on protein structure, stability and Ca²⁺-binding function.

In summary, while the paper demonstrates promise in addressing dysferlin deficiency using CRISPR-mediated gene editing, it falls short in various critical aspects. Addressing these limitations is vital to ensure the safety, efficacy, and long-term viability of this therapeutic approach before clinical translation.

Minor points:

- Line 38 and others: LGMD2B is now known as LGMDR2 please consider this change. We now mention the new terminology (line 39).
- Line 42: tandem C2: I can't find a reference to explain why this C2 domain are called tandem? do you have one pertinent reference for this? On line 61, even though dysferlin may not inherently possess redundant domains, existing references indicate that certain domains of dysferlin are deemed less critical. Refs: doi:10.1016/j.ymthe.2017.05.013; doi: 10.1126/scitranslmed.3000951 for example. The closest reference to our knowledge is Lek et al., 2010, BMC Evol Biol 10, doi: 10.1186/1471-2148-10-231. However, we acknowledge that the term "tandem" is often used to refer to dysferlin's C2 domains without further explanation and hence have decided to remove it.
- Line 163: two spaces between We generated. This has been corrected.
- Line 189: the reticular pattern is not clearly seen in the image provided. We now provide a new image where the reticular pattern can be seen more clearly (Suppl. Fig 15b).
- Line 217: It seems somewhat strange that these authors do not acknowledge the potential for exon skipping in dysferlin, especially considering their publication using this technique and the number of articles demonstrating its efficacy. Refs: doi: 10.1007/978-1-0716-2772-3_11; doi: 10.1089/nat.2019.0788; doi: 10.1016/j.omtn.2018.08.013; doi: 10.3233/JND-150109; 10.1093/hmg/ddv141. We appreciate that the reviewer points to our previous research. We added the corresponding reference (line 249 of the revised manuscript with marked changes).
- Line 222: The database suggests that only 38 patients might benefit from this approach, indicating a limited number of potential beneficiaries. Dysferlin-deficient muscular dystrophy, like all other muscular dystrophies, is a rare disease. Moreover, no mutational hotspots are described in the DYSF gene. There are only a few founder mutations that are more prevalent and have a high carrier frequency in certain population subsets, c.4872_4876delinsCCCC being one of them. This is mentioned in the introduction.
- Line 230: the mutation V67D in the C2A is also quite well explored on its effect on protein structure and ions binding. We have added a reference (line 261 of the revised manuscript with marked changes).
- Line 232: The systematic assessment of splicing defects caused by missense mutations has been thoroughly investigated in the past see: doi: 10.1002/humu.22710. The reference has been added (line 266 of the revised manuscript with marked changes).
- Line 313: I was unable to locate the specific experiment addressing clonal drift. We show that the overall percentage and proportion of different types of indels resulting from different DNA repair outcomes at the target site remains constant in all replicates of edited MuSC from both patients between day 4 (Fig. 3d) and day 8 (Supplementary Fig. 5) post-editing. The fact that the allele frequencies of the edited cell populations did not change over time indeed strongly suggests an absence of clonal drift during the observation period. Long-term monitoring of the clonality of ex vivo expanded and in vivo engrafting gene-edited MuSC populations is a very interesting future prospect (please see lines 365-371 of the revised manuscript with marked changes).
- Line 456: There is no experiment provided that demonstrates time-dependent membrane repair. We have corrected this.
- Line 493: it is not clear if the MuSC are obtained from 4- to 12-week -old homozygous Dysf hEx44wt/hEx44wt or from sibling line 513? MuSC were isolated from both homozygous Dysf<hEx44wt/hEx44wt> and <hEx44mut/hEx44mut> mice. MuSC from homozygous hEx44wt mice were utilized as positive control for dysferlin expression analysis (Fig. 7d-f). For

transplantation, MuSC were isolated from homozygous hEx44mut mice and gene-edited, and siblings were used as MuSC donors whenever possible (now more clearly written in the materials and methods, *lines 613-615 of the revised manuscript with marked changes*).

- Line 544: The statistical methods employed in this study may not be entirely suitable due to the notably small group sizes. For instance, using mean and standard deviation might not adequately represent the data due to its limited sample size. *We always show the individual data points, and there is no measurement where the data distribution appears non-gaussian.*

Reviewer #3 (Remarks to the Author):

In the paper titled "Gene-edited primary muscle stem cells rescue dysferlin-deficient muscular dystrophy" by Escobar et al., the authors successfully showcase the restoration of dysferlin function in cells afflicted with a prevalent mutation in the DYSF gene through gene editing. Despite the mutation-specific nature of the strategy, the study affirms its efficacy in both in vitro and in vivo settings, particularly in a humanized model of the disease. The paper is commendably written, with clear presentation of data. My primary suggestion pertains to a substantial comment regarding the in vivo quantification of the recovery, along with a few minor suggestions.

Major:

In the section titled "Exon 44 re-framed murine MuSC regenerate muscle and rescue dysferlin in hEx44mut mice," I recommend the inclusion of a graph depicting the count of positive myofibers alongside the quantification of restoration efficiency through western blot analysis. This addition would significantly enhance the clarity of the differences observed among the three pretreatment regimens and provide a more comprehensive understanding of the overall in vivo efficacy of the proposed strategy.

We now provide a graph depicting the count of dysferlin positive fibers (Fig. 8b). We have also performed a Western blot quantification:

For most grafted muscles, dysferlin rescue is below the detection threshold using Western blot, which can be explained by the rather small size of the graft, compared with the whole host muscle. This is likely a limitation of the donor cells growing only in close proximity to the injection site/needle trajectory and could be improved by performing multiple injections instead of a single injection. We would prefer to not include the Western blot in the revised manuscript but are happy to do so if this is the reviewer's recommendation.

Minor:

- Line 111: reference for Fig 2b is missing in the main text. *The reference has been added.*
- Line 114: define DSB. *The abbreviation has been added in the introduction (line 52).*
- SpCas9 is sometimes in italics and sometimes not: please be consistent. *Now corrected.*

- Fig 2g: quantify the dysferlin recovery in the 2 clones compared with the hiPSCs controls. We now provide a densitometric analysis (Supplementary Fig. 2b).
- Line 164: reference for Fig 6a is missing in the main text. The reference has been added.
- Discussion: the long-term in vivo effect of the proposed treatment is not discussed. Please provide the authors' view about this topic. We have now added this to the discussion (lines 320-333/343 of the revised manuscript with marked changes).